# Improved identification accuracy in equation learning via comprehensive $R^2$-elimination and Bayesian model selection

**Daniel Nickelsen**                                                                                 *danieln@aims.ac.za*
*Computational Statistics and Data Analysis (CSDA), Institute for Mathematics, University of Augsburg, Germany*
*African Institute for Mathematical Sciences (AIMS), Cape Town, South Africa*

**Bubacarr Bah**                                                                           *bubacarr.bah1@lshtm.ac.uk*
*Medical Research Council Unit The Gambia, London School of Hygiene & Tropical Medicine, The Gambia*
*African Institute for Mathematical Sciences (AIMS), Cape Town, South Africa*

**Reviewed on OpenReview:** *https://openreview.net/forum?id=0ck7hJ8EVC*

## Abstract

In the field of equation learning, exhaustively considering all possible equations derived from a basis function dictionary is infeasible. Sparse regression and greedy algorithms have emerged as popular approaches to tackle this challenge. However, the presence of multicollinearity poses difficulties for sparse regression techniques, and greedy steps may inadvertently exclude terms of the true equation, leading to reduced identification accuracy. In this article, we present an approach that strikes a balance between comprehensiveness and efficiency in equation learning. Inspired by stepwise regression, our approach combines the coefficient of determination, $R^2$, and the Bayesian model evidence, $p(\boldsymbol{y}|\mathcal{M})$, in a novel way. Our procedure is characterized by an comprehensive search with just a minor reduction of the model space at each iteration step. With two flavors of our approach and the adoption of $p(\boldsymbol{y}|\mathcal{M})$ for bi-directional stepwise regression, we present a total of three new avenues for equation learning. Through three extensive numerical experiments involving random polynomials and dynamical systems, we compare our approach against four state-of-the-art methods and two standard approaches. The results demonstrate that our comprehensive search approach surpasses all other methods in terms of identification accuracy. In particular, the second flavor of our approach establishes an efficient overfitting penalty solely based on $R^2$, which achieves highest rates of exact equation recovery.

## 1 Introduction

Uncovering the underlying laws governing a system is essential for understanding its behavior, and mathematical equations serve as a concise representation of these laws. Equipped with such equations, predictions can be made, and valuable insights can be derived analytically. The pursuit of inferring these governing equations directly from observations has a long history, dating back to Johannes Kepler's deduction of planetary motion laws in 1609. In modern times, significant progress has been made in automated equation inference using machine learning techniques, a discipline commonly referred to as *equation learning* or *symbolic regression*.

In contrast to deep learning, equation learning focuses on maximizing model expressivity while minimizing complexity to ensure interpretability. This delicate balance between interpretability and expressivity constitutes the central challenge of equation learning. Other challenges include stable feature selection, solvability of learnt equations, computational feasibility, and small data. Greedy algorithms and regularization techniques applied to regression models built from basis functions are often employed to address these challenges.

Greedy algorithms, like stepwise regression and iterative thresholding, build or reduce the model space by iterative inclusion or exclusion of terms without reconsideration at later steps. Although these methods are computationally efficient, they come at the price of potentially excluding the true model from consideration.

On the other hand, regularization transforms the search into an optimization problem, where all candidate models serve as points in the objective function landscape. However, finding the one global minimum that corresponds to the true model is not guaranteed, and local minima may only accidentally lead to the true model. Multicollinearity among basis functions further complicates optimization algorithms due to the jagged nature of the optimization space.

In this work, we tackle the aforementioned challenges of equation learning by enhancing the stepwise regression approach by a stage wise comprehensive model search with only a minor reduction of model space. Our method begins with an elimination process that employs the computationally inexpensive coefficient of determination, $R^2$, in order to assess almost all individual candidate models belonging to a complexity class. Selecting the best model from that process enables subsequent model selection based on the Bayesian model evidence. The model evidence reflects the probability of a model being the true model for the given data, making it an ideal criterion for penalizing overfitting and addressing the central challenge of equation learning: achieving the optimal balance between interpretability and expressivity. By employing specifically tuned conjugate priors within an empirical Bayes framework, we circumvent the computationally demanding estimation of the evidence and instead derive it analytically.

To evaluate our approach, we employ artificially generated data from known models. This choice of evaluation allows us to assess whether our strategy can uncover the ground truth model, serving as a testament to its ability to strike the delicate balance between expressivity and interpretability in realistic scenarios. We compare our method against two standard techniques, least absolute shrinkage and selection operator (LASSO) and least-angle regression (LARS), as well as four state-of-the-art methods available in the `PySINDy` package: sparse relaxed regression (SR3), forward regression orthogonal least squares (FROLS), sequentially thresholded least squares (STLSQ), and best subset selection via mixed-integer optimized sparse regression (MIOSR). Additionally, we propose bi-directional stepwise regression equipped with the Bayesian model evidence (BSR). We evaluate the identification accuracy of approximately 80 scenarios for each system and assess the forecasting accuracy based on 100 initial values for each scenario and system, amounting to a total of around 8,000 tests. Our findings demonstrate that our proposed methods outperform other approaches in terms of identifying the correct model and can achieve competitive forecasting accuracy.

Our paper is organized as follows. We begin with a short overview of existing method in Sec. 2, and introduce our approach and the methods we compare to in Sec. 3. In Sec. 4 we present our numerical results, which we discuss in Sec. 5 and conclude in Sec. 6. A largely self-contained description of our approach with more details and background is provided in the appendix. Our code is available in the Supplemental Material.

## 2 Related work

The literature on equation learning can be roughly divided into three approaches: evolutionary algorithms, tree and neural network representations, and regression. A well-known example of evolutionary algorithms is EUREQA, a commercial software used for symbolic regression (Dubčáková, 2011; Stoutemyer, 2013). A recent open source alternative to EUREQA also using evolutionary algorithms is the python package PYSR (Cranmer, 2023). Tree representations construct equations by combining basic operations and are then employed to minimize regularized objective functions (Vaddireddy et al., 2020), using posterior sampling with sparsity-promoting priors (Jin et al., 2019), or through mixed-integer linear programming (Neumann et al., 2020). Similarly, neural network architectures have been used to represent equations, where network nodes are replaced by expression building blocks (Martius & Lampert, 2016; Sahoo et al., 2018; Werner et al., 2021). These neural networks are trained with a regularized minimizer applied to objective functions (Rackauckas et al., 2020). Another approach, which allows for the incorporation of physics-informed properties such as symmetries, has also been developed (Udrescu & Tegmark, 2020).

These methods find applications in complementing deep neural networks to enhance generalization (Arabshahi et al., 2018) and reducing the data requirement for training (Yang et al., 2021). They are also utilized in uncovering complex ecosystem dynamics (Chen et al., 2019).

The above approaches are highly non-linear and typically involve complex optimization algorithms. A simpler approach is based on linear regression models, where features are replaced by basis functions derived from

observed data. Regularized regression ensures sparse weight estimates on the basis functions (Hastie et al., 2009), resulting in concise mathematical expressions. One prominent and widely used method for sparse regression is the LASSO with $\ell_1$-regularization (Tibshirani, 1996). The advantage of $\ell_1$-regularization is the convexity of the objective functions, which allows for efficient optimization. However, as we will discuss later, while sparse regression performs well with independent features in reconstruction tasks, the correlations introduced by basis function expansion can lead to detrimental instability in equation learning (Su et al., 2017; Rudy et al., 2017). Recent advancements of LASSO, such as SR3, can be found in Zheng et al. (2019); Tibshirani & Friedman (2020).

Greedy algorithms like stepwise regression, FROLS and STLSQ, as well as sparse relaxed regression (i.e. SR3) have proven to be more successful than LASSO in equation learning (Champion et al., 2020; Kaiser et al., 2018). The latter three algorithms (FROLS, STLSQ, SR3) are implemented in the PYTHON package PySINDy, which is designed for the sparse identification of nonlinear dynamics (SINDy) (Brunton et al., 2016). SINDy, initially using STLSQ, has seen numerous extensions, including applications to partial differential equations (Rudy et al., 2017), improved noise robustness through automated differentiation (Kaheman et al., 2020) and the weak form of differential equations (Messenger & Bortz, 2021a;b), supplemented with a-posteriori (MAP) estimates (Niven et al., 2020), re-weighted $\ell_1$-regularization (Cortiella et al., 2021), relaxed regularization (Champion et al., 2020), and most recently, best subset selection using MIOSR (Bertsimas & Gurnee, 2023). In Nardini et al. (2020), spatio-temporal biological data is first denoised using artificial neural network before partial differential equations are learnt from the processed data using STLSQ.

In Bayesian linear regression, sparsity-promoting priors are used instead of regularization for equation learning (Nayek et al., 2021). Thresholded sparse regression has been translated into a Bayesian formulation with implied error bars in Zhang & Lin (2018; 2021). By restricting basis functions to quadratic order, the linear structure of the models allows for deterministic results even in the case of the more challenging $\ell_0$-regularization (Schaeffer et al., 2018).

## 3 Equation learning as a regression problem

The goal of equation learning is to find a function $f(\boldsymbol{x})$ that accurately represents the relationship between the input variables $\boldsymbol{x}$ and the output variable $y$. In the context of regression models, we consider a dataset consisting of $N$ observations, where the inputs are organized in a design matrix $\boldsymbol{X}$ and the corresponding outputs are modeled by a random variable $Y$. We can express the relationship between $\boldsymbol{x}$ and $y$ as follows:

$$Y = f(\boldsymbol{x}) + \sigma Z, \tag{1}$$

where the feature vector $\boldsymbol{x}$ is a row of $\boldsymbol{X}$, $Z$ is a standard normal random variable, and $\sigma^2$ is the variance of the noise term. Our goal is to represent the unknown function $f(\boldsymbol{x})$ using a basis function expansion, i.e. as a linear combination of $p$ basis functions $k_n(\boldsymbol{x})$,

$$f(\boldsymbol{x}) = \sum_{n=1}^{p} w_n k_n(\boldsymbol{x}), \tag{2}$$

where $w_n$ denotes the weights associated with each basis function. Common choices for the basis functions $k_n(\boldsymbol{x})$ involve products and powers of the individual features $x_j$. For example, we can have $k_1(\boldsymbol{x}) = x_1^2$, $k_2(\boldsymbol{x}) = x_1 x_2$, $k_3(\boldsymbol{x}) = x_1 x_3$, and build $f(\boldsymbol{x}) = w_1 x_1^2 + w_2 x_1 x_2 + w_3 x_1 x_3$.

Constructing a basis function matrix $\boldsymbol{K}$, with elements $K_{in} = k_n(\boldsymbol{x}_i)$, the ordinary least squares (OLS) estimates for the weights $w_n$ are given by Montgomery et al. (2012)

$$\hat{\boldsymbol{w}} = (\boldsymbol{K}^\mathsf{T} \boldsymbol{K})^{-1} \boldsymbol{K}^\mathsf{T} \boldsymbol{y}, \tag{3}$$

where $\boldsymbol{y}$ represents a vector consisting of $N$ samples of the target variable $Y$. With the estimated weights $\hat{\boldsymbol{w}}$, we can make predictions $\hat{\boldsymbol{y}} = \boldsymbol{K}\hat{\boldsymbol{w}}$.

Equation learning involves finding a small subset of $k_n(\boldsymbol{x})$ for which the corresponding weights $w_n$ are estimated to fit the data, while the remaining weights can be set to zero without compromising expressivity.

The challenging part lies indeed in the selection of the appropriate basis functions $k_n(\boldsymbol{x})$, after that, in view of Eq. (3), the actual learning of the model is straightforward. On one hand, we require the model to have enough flexibility (expressivity) to minimize bias, ensuring that it captures the underlying relationship between $\boldsymbol{x}$ and $Y$. On the other hand, we want to avoid overfitting, which can lead to high variance in predictions. This trade-off between bias and variance guides the determination of the number of non-zero weights, denoted as $m$, or equivalently, the model size. In addition to finding the appropriate model size, we also aim to identify the "correct" set of basis functions $k_n(\boldsymbol{x})$, which corresponds to recovering the true $f(\boldsymbol{x})$ from data generated by Eq. (1).

## 3.1 Model class

Apart from the equations that can directly be written in the form of Eq. (2), a prominent application of equation learning is the learning of dynamical systems,

$$\dot{\boldsymbol{x}}(t) = f(\boldsymbol{x}(t)), \tag{4}$$

where $\dot{\boldsymbol{x}}(t)$ denotes the time derivative of $\boldsymbol{x}(t)$. To map this problem to Eq. (2), the response variable can be computed from finite differences, i.e. $y_i = \frac{x_{i+1} - x_i}{\Delta t}$ for each component of $\boldsymbol{x}$ and a fixed time step $\Delta t$.

It must be noted that the assumption of normally distributed $Y$ in Eq. (1) might not be justified in the context of dynamical system learning due to temporal correlations. On the other hand, if the error structure in $\boldsymbol{x}(t)$ is normally distributed, then the differences $x_{i+1} - x_i$ will again be normally distributed, even in the presence of correlations. However, a time varying noise structure would cause heteroscedasticity which could impede statistical methods. In a different approach, the weak form of the differential equation (4) can be utilized; we refer the reader to Messenger & Bortz (2021a) for details.

A restriction for the regression models to stay linear in its parameters is that parameters of basis functions may only enter as weights $w$. Basis functions like $e^{ax}$, $\ln(a + x)$, $\cos ax$, $x^a$, $\frac{1}{(a+x)^m}$, ... with internal parameter $a$ are not suitable.

This restriction might seem quite limiting. On the other hand, the function $f(\boldsymbol{x}(t), \boldsymbol{w})$ defining a dynamical system typically is linear in its parameters $\boldsymbol{w}$. The reason for that is that these functions often reproduce when differentiated, which can be used to eliminate these functions, retrieving the standard linear form in Eq. (2). Many special functions like Bessel, Hankel, Struve and Meijer functions are even defined as solutions of differential equations linear in their coefficients. In general, by considering the differentiated response variable, $y \mapsto \frac{dy}{dx} \simeq \frac{y(x_{i+1}) - y(x_i)}{x_{i+1} - x_i}$, if necessary to higher order, we can learn a surprisingly broad class of equations relating $y$ and $\boldsymbol{x}$, even relations that do not exist in closed form.

## 3.2 Regularized regression

The OLS estimates (3) would always involve all terms $k_n(\boldsymbol{x})$ of the basis function dictionary. Equation learning hence is directly related to sparse regression in this context. A common way to mitigate overfitting and thus promote sparsity in regression is to use regularization,

$$\hat{\boldsymbol{w}} = \arg\min_{\boldsymbol{w} \in \mathbb{R}^p} \left[ \left\| \boldsymbol{K}\boldsymbol{w} - \boldsymbol{y} \right\|_2^2 + \lambda \|\boldsymbol{w}\|_q \right], \tag{5}$$

where $\|\boldsymbol{w}\|_q = [\sum_n |w_n|^q]^{1/q}$, and $\lambda$ is the Lagrange parameter that sets the strength of the $\ell_q$ penalty. The standard, sparsity promoting choice is $q = 1$ which is known as the LASSO (Tibshirani, 1996). The choice $q = 0$, for which $\|\boldsymbol{w}\|_0 = \sum_n \delta_{w_n,0}$ is the number of non-zero weight estimates, is often called *best subset selection* and requires specialized optimization algorithms (Zhu et al., 2020; Hastie et al., 2020), like MIOSR (Bertsimas & Gurnee, 2023).

A relaxed regularization like SR3 can be introduced by letting the penalty act on an auxiliary variable $\boldsymbol{u}$, where distance of $\boldsymbol{u}$ to the actual weights is controlled by an additional regularization term, e.g. $\|\boldsymbol{w} - \boldsymbol{u}\|_q$ (Zheng et al., 2019).

### 3.3 Model selection

Taking a different route, sparse solutions of Eq. (3) may be realized by model selection, where basis functions are not discarded by shrinkage of their weights, instead a criterion is used to select a model from a set of candidate models. Here, a candidate model $\mathcal{M}$ is defined via a subset of $m$ basis functions $k_1(\boldsymbol{x}), ..., k_m(\boldsymbol{x})$, which mathematically translates into building a $N \times m$ submatrix $\boldsymbol{K}'$ from the full $N \times p$ basis function matrix $\boldsymbol{K}$. To cope with the huge candidate model space, equation learning utilizing model selection typically involves greedy iterations in which the model space is reduced drastically at each iteration.

Selection based equation learning differ in the strategy to trawl through model space and by the choice of selection criteria. A computationally cheap criterion would be the coefficient of determination,

$$R^2 = \frac{\boldsymbol{y}^{\mathrm{T}} \boldsymbol{K}' (\boldsymbol{K}'^{\mathrm{T}} \boldsymbol{K}')^{-1} \boldsymbol{K}'^{\mathrm{T}} \boldsymbol{y}}{\boldsymbol{y}^{\mathrm{T}} \boldsymbol{y}}, \tag{6}$$

here in a simplified form for standardized $Y$ (Montgomery et al., 2012). However, $R^2$ is known for its lack of overfitting penalty as it would just increase with decreasing sparsity.

Alternatives like adjusted versions of $R^2$ exist to incorporate an overfitting penalty, but here we make use of the Bayesian model evidence

$$p(\mathcal{M}) \propto p(\boldsymbol{y}|\mathcal{M}) = \int \mathrm{d}\sigma \int \mathrm{d}\boldsymbol{w} \, p_{\mathrm{li}}(\boldsymbol{y}|\boldsymbol{w}, \sigma, \mathcal{M}) \, p_{\mathrm{pr}}(\boldsymbol{w}, \sigma), \tag{7}$$

where $\mathcal{M}$ defines a normal likelihood function $p_{\mathrm{li}}(\boldsymbol{y}|\boldsymbol{w}, \sigma, \mathcal{M})$ via Eq. (1) substituting $\boldsymbol{K}'$ for $\boldsymbol{K}$ in Eq. (2). For a conjugate prior $p_{\mathrm{pr}}(\boldsymbol{w}, \sigma)$, the marginalization above can be done analytically and $p(\boldsymbol{y}|\mathcal{M})$ is known exactly, as detailed in App. B.

The use of $p(\boldsymbol{y}|\mathcal{M})$ is often motivated by its excellent overfitting penalizing properties (Occam's razor), which can be ascribed to $p(\boldsymbol{y}|\mathcal{M})$ being proportional to the probability $p(\mathcal{M})$ of the model $\mathcal{M}$ being the true model for the data $(\boldsymbol{y}, \boldsymbol{X})$ (Murphy, 2012). The downside of using $p(\boldsymbol{y}|\mathcal{M})$ is the imperative to specify $p_{\mathrm{pr}}(\boldsymbol{w}, \sigma)$ even in cases of scarce prior knowledge, which can have a significant impact on $p(\boldsymbol{y}|\mathcal{M})$. In an empirical Bayes approach, we fix $p_{\mathrm{pr}}(\boldsymbol{w}, \sigma)$ by exploiting the normality property of linear regression which implies that estimates $\hat{\boldsymbol{w}}$ are normally distributed with the mean given by the true values of $\boldsymbol{w}$ and the variance given by

$$\hat{\sigma}^2 = \frac{(\boldsymbol{y} - \hat{\boldsymbol{y}})^{\mathrm{T}}(\boldsymbol{y} - \hat{\boldsymbol{y}})}{N - p}. \tag{8}$$

The details of this procedure is included in App. A.

### 3.4 Stepwise regression

A standard greedy algorithm to build $\boldsymbol{K}'$ is stepwise regression, where terms are added or removed from the model step by step based on a criterion (Montgomery et al., 2012). Here, we promote using the Bayesian model evidence $p(\boldsymbol{y}|\mathcal{M})$ in Eq. (7) as criterion, which is rarely used due to its intricacies in terms of prior selection and computational cost (Hastie et al., 2009). Our Bayesian stepwise regression (BSR) procedure is bi-directional, that is, we start with an empty model and add the $k_n(\boldsymbol{x})$ that maximizes $p(\boldsymbol{y}|\mathcal{M})$, add a second $k_n(\boldsymbol{x})$ to $\boldsymbol{K}'$ maximizing $p(\boldsymbol{y}|\mathcal{M})$, and so on (forward selection). Once $p(\boldsymbol{y}|\mathcal{M})$ cannot be increased further by adding more $k_n(\boldsymbol{x})$ to $\boldsymbol{K}'$, we start removing columns from $\boldsymbol{K}'$ in the same fashion until again $p(\boldsymbol{y}|\mathcal{M})$ is maximized (backward selection). We continue forward and backward selection until $p(\boldsymbol{y}|\mathcal{M})$ cannot be increased in either selection direction. From including the backward direction and due to the overfitting penalty of $p(\boldsymbol{y}|\mathcal{M})$ we expect our procedure to be more parsimonious in terms of model size.

Other stepwise regression procedures are FROLS, LARS and STLSQ. In FROLS, forward selection is applied with the correlation to the target variable as criterion such that Eq. (5) with $q = 2$ (Ridge regression) is minimized (Billings, 2013). A related stepwise algorithm is least-angle regression (LARS) where the correlation between $k_n(\boldsymbol{x})$ and residuals is used to build the model in a forward procedure. Following a backward thresholding procedure, STLSQ starts with the full model and alternates between Ridge regression and removing terms $k_n(\boldsymbol{x})$ with weights $w_n$ below a pre-defined threshold (Brunton et al., 2016). The STLSQ procedure is the standard method in `PySINDy`.

### 3.5 Comprehensive search (CS)

An exhaustive search algorithm would consider all $\sum_{m=1}^{p} \binom{p}{m} = 2^p - 1$ combinations of terms in the model equation which obviously quickly becomes computationally infeasible. However, since for equation learning we are interested in parsimonious models, we may choose a small candidate model size $m$ and explore all $\binom{p}{m}$ possible models $\mathcal{M}$ defined by $\boldsymbol{K'}$ within that budget. For a fixed model size $m$, overfitting penalty is not important, and we may use $R^2$ which is particularly cheap to compute for standardized data, c.f. Eq. (6). In this way, we can single out the best models in terms of $R^2$ for different budgets $m$.

To find the best model size $m$, we utilize the overfitting penalty property of the model evidence $p(\boldsymbol{y}|\mathcal{M})$. Starting with $m = 1$, we compute $p(\boldsymbol{y}|\mathcal{M})$ for a fixed number $s = p/2$ of best models selected by $R^2$, and continue to do so for increasing $m$ until a stopping criterion is reached. After that, we select the model that maximizes $p(\boldsymbol{y}|\mathcal{M})$ out of the top models selected by $R^2$ across all considered model sizes. We refer to this method as CS-$p(\mathcal{M})$.

---

**Algorithm 1** Generate dictionary of basis functions to produce matrix $\boldsymbol{K}$

---

1: Function FuncDict($\boldsymbol{X}$)
2: **Input:** data $\boldsymbol{X}$ with $N$ datapoints, $l$ features
3: **Parameters:** maximum degree $M_1$ for individual features, maximum degree $M_2$ for term
4: build all powers $(X_{ij})^{m_j}$, $m_j = (1, ..., M_1)$         ▷ pre-computed for speed, limited to power $M_1$
5: initialize counter $p = 0$
6: **for** all unique $l$-tuples $(m_1, ..., m_l)$ **do**
7:      **if** $\sum_j m_j \leq M_2$ **then**         ▷ ensure that collective power is limited to $M_2$
8:         $p := p + 1$
9:         $\boldsymbol{K}_{:,p} := \prod_j \boldsymbol{X}_{:,j}^{m_j}$         ▷ basis function matrix, candidate models in columns
10:      **end if**
11: **end for**
12: **Return:** basis function matrix $\boldsymbol{K}$, shape $N \times p$

---

**Algorithm 2** Model ranking using $R^2$

---

1: Function TopRsq($\boldsymbol{y}, \boldsymbol{K}, m$)
2: **Input:** response data $\boldsymbol{y}$, basis function matrix $\boldsymbol{K}$, number $m$ of terms for candidate models
3: **Parameters:** number $t = 25$ of top models
4: initialize criterion $\boldsymbol{c}$         ▷ flexible length, to store $R^2$ for candidate models
5: initialize model indices $\boldsymbol{M}$         ▷ $p$ rows indicating terms part of models, flexible number of columns
6: initialize model number $i = 0$
7: **for** all $\boldsymbol{n} = (n_1, ..., n_p) \in \{0, 1\}^p$, $\sum_j n_j = m$ **do**         ▷ $\binom{p}{m}$ possible selections of terms for fixed size $m$
8:      $i := i + 1$,
9:      reduce $\boldsymbol{K'} := \boldsymbol{K}_{:,\boldsymbol{n}}$         ▷ extract terms $\boldsymbol{n}$ to define candidate model via $\boldsymbol{K'}$
10:      determine $R^2$ for $\boldsymbol{K'}$ and $\boldsymbol{y}$ from Eq. (6)
11:      store $c_i := R^2$ and $\boldsymbol{M}_{:,i} := \boldsymbol{n}$         ▷ each column of $\boldsymbol{M}$ indicates a model with $R^2$ value $c_i$
12: **end for**
13: sort columns of $\boldsymbol{M}$ and $\boldsymbol{c}$ by $\boldsymbol{c}$ (descending)         ▷ first columns of $\boldsymbol{M}$ now indicate top models in terms of $R^2$
14: **Return:** top models $\boldsymbol{M}_{:,:t}$ (shape $p \times t$), criterion $\boldsymbol{c}_{:t}$ (length $t$)         ▷ only return the $t$ best models

---

As a stopping criterion, we may use the first decrease of $p(\boldsymbol{y}|\mathcal{M})$, but we propose a criterion purely based on $R^2$ which proved superior in our study. The $R^2$ criterion we propose builds on the observation that true terms $k_n(\boldsymbol{x})$ have a tendency to be consistently selected in the top $R^2$ models. We therefore keep incrementing $m$ until no new $k_n(\boldsymbol{x})$ is selected consistently, and then build the inferred model from those consistently selected terms. While this is a empirically developed method, it is motivated by the reasonable assumption that the maximization of $R^2$ is dominated by the true $k_n(\boldsymbol{x})$. Therefore, as soon as we look at the $R^2$ values of models one term larger than the true model, the procedure is forced to randomly select one extra term in addition to the consistently selected true terms. We refer to this method solely based on $R^2$ as CS-$R^2$.

Since $\binom{p}{m}$ still becomes very large for larger model sizes $m$, we add an additional pruning step, in which all $k_n(\boldsymbol{x})$ that have consistently not been selected for two subsequent iterations are removed from the basis function dictionary for all following iterations. The stopping criterion and the pruning step may classify

---

**Algorithm 3** Comprehensive search with $R^2$ and $p(\boldsymbol{y}|\mathcal{M})$

---

1: Function CHS($\boldsymbol{y}, \boldsymbol{X}$)
2: **Input:** data $\boldsymbol{y}, \boldsymbol{X}$
3: **Parameters:** maximum number $m_{\max}=8$ of terms, number $s=\frac{p}{2}$ of top models, threshold $c_{\min}=0.75$ for term selection
4: $\boldsymbol{K}, p := \text{FuncDict}(\boldsymbol{X})$
5: initialize feature rating $\boldsymbol{F}$ of shape $p \times m_{\max}$               ▷ rate importance of terms for different model sizes $m$
6: initialize $search := True$ and number of terms $m := 0$
7: **while** $search$ and $m \leq m_{\max}$ **do**        ▷ increment model size $m$ in each iteration until stopping criterion reached
8:      $m := m + 1$
9:      $\boldsymbol{M}, \boldsymbol{c} := \text{TopRsq}(\boldsymbol{y}, \boldsymbol{K}, m)$         ▷ obtain list of models and with their $R^2$ values for fixed model size $m$
10:      append $\boldsymbol{M}$ to $\boldsymbol{M}_{\text{all}}$         ▷ append models to index matrix keeping models for all $m$
11:      $\boldsymbol{F}_{:,m} := \sum_j^s c_j \boldsymbol{M}_{:,j}$      ▷ counts how often terms are selected in $s$ top models, weighted by $R^2$ criterion
12:      normalize $\boldsymbol{F}_{:,r} := \boldsymbol{F}_{:,r} / \max(\boldsymbol{F}_{:,r})$         ▷ to have ratings between 0 and 1
13:      **if** $r \geq 2$ **then**
14:          index $\boldsymbol{i}_0 := (\boldsymbol{F}_{:,r} + \boldsymbol{F}_{:,r-1} = 0)$         ▷ $i_0$=True if terms not selected for two successive model sizes
15:          remove $\boldsymbol{K}_{:,i_0}$ from $\boldsymbol{K}$         ▷ reduce model equation space by those terms
16:          index $\boldsymbol{i}_1 := (\boldsymbol{F}_{:,r} \geq c_{\min})$         ▷ indexes terms with significant rating across best $s$ models of size $r$
17:          index $\boldsymbol{i}_2 := (\boldsymbol{F}_{:,r-1} \geq c_{\min})$         ▷ same for previously considered model size $r-1$
18:          **if** $\boldsymbol{i}_1 = \boldsymbol{i}_2$ **then**
19:             $search := False$         ▷ if term is selected twice in a row like this, conclude search
20:             $\boldsymbol{M}^* := \boldsymbol{i}_1, \boldsymbol{K}^* := \boldsymbol{K}_{:,\boldsymbol{M}^*}$         ▷ defines model $\mathcal{M}^*$ learnt by CS-$R^2$
21:          **end if**
22:      **end if**
23: **end while**
24: Compute $p(\boldsymbol{y}|\mathcal{M})$ for models $\boldsymbol{M}_{\text{all}}$ using Eq. (7), store $\boldsymbol{M}^{(1)}$ maximizing $p(\boldsymbol{y}|\mathcal{M})$    ▷ defines model $\mathcal{M}^{(1)}$ learnt by CS-$p(\mathcal{M})$
25: Obtain OLS estimates $\hat{\boldsymbol{w}}^*$ for $\boldsymbol{K}^*$ and $\hat{\boldsymbol{w}}^{(1)}$ for $\boldsymbol{K}^{(1)} := \boldsymbol{K}_{:,\boldsymbol{M}^{(1)}}$ using Eq. (3)

26: **Return:** $\boldsymbol{K}^*$ with $\hat{\boldsymbol{w}}^*$ (model inferred by CS-$R^2$), $\boldsymbol{K}^{(1)}$ with $\hat{\boldsymbol{w}}^{(1)}$ (model inferred by CS-$p(\boldsymbol{y}|\mathcal{M})$)

---

our procedure as a greedy algorithm. However, due to the comprehensive search for each considered $m$, we consider a drastically larger model space than other greedy algorithms. It is also worth noting that our procedure can be extended to cases where $p(\boldsymbol{y}|\mathcal{M})$ needs to be estimated by computationally costly evidence estimators, as we effectively reduce the pool of candidate models to just a few.

The near-exponential increase of $\binom{p}{m}$ for $m \ll p$ naturally raises the questions of how our model scales with candidate model size $m$. On the other scales the computation of $R^2$ almost linearly in $N$ and $m$. Given the fact that in equation learning the goal is to find a minimal model (in fact, models considered in the literature hardly ever exceed $m = 4$ terms), the runtime per iteration in $m$ stays well below 1 min on a standard laptop. A detailed analysis is provided in App. C and Fig. 7.

We summarized details of our CS approach in Alg. 1-3. Illustrations and more details can be found in App. C.

## 4 Numerical experiments

We compare all introduced methods (LASSO, LARS, CS-$R^2$, CS-$p(\mathcal{M})$, BSR, SR3, FROLS, STLSQ, MIOSR) in three numerical experiment. For LASSO and LARS, we use the PYTHON package `scikit-learn` (Pedregosa et al., 2011), for CS-$R^2$, CS-$p(\mathcal{M})$ and BSR we use our own implementation, and for SR3, FROLS, STLSQ, MIOSR we use the PYTHON package `PySINDy` (de Silva et al., 2020; Kaptanoglu et al., 2022). As mixed-integer optimizer for MIOSR we used GUROBI with an academic license (Gurobi Optimization, LLC, 2023). For all methods from `scikit-learn` and `PySINDy` we used 5-fold cross validation and 3 refinement steps to determine optimal hyperparameters for each application separately. Our own methods, CS-$R^2$ and CS-$p(\mathcal{M})$, are not very sensitive to hyperparameters and we worked out the universally best values which, in contrast to the methods we compare to, were used for all applications. The values of the hyperparameters are included in Alg. 1-3. The BSR method we implemented comes without hyperparameters. Tab. 1 gives an overview over all considered methods.

In all experiments, the data is artificially generated, with the advantage that we know the true model. In the first experiment, we assess the identification accuracy of 100 random polynomials. Since `PySINDy` is tailored to learning dynamical systems and cannot directly be applied to learning polynomials, we omitted those

Table 1: Equation learning techniques used in this study building on regression models as in Eqs. (1)-(2).

| Acronym | Full name | Description | Reference |
|---|---|---|---|
| LASSO | Least absolute shrinkage selection operator | Regularized regression as in Eq. (5) for $q=1$. | (Tibshirani, 1996). |
| LARS | Least angle regression | Forward stepwise regression using correlations between basis functions $k_n$ and residuals. | (Efron et al., 2004). |
| CS-$R^2$ | Comprehensive search $R^2$ elimination | For an increasing model size $m$, the $R^2$ score is calculated for all possible models. Based on the number of times basis functions $k_n(\boldsymbol{x})$ contribute to the models with largest $R^2$, terms $k_n(\boldsymbol{x})$ are rated and the lowest rated $k_n(\boldsymbol{x})$ are excluded. Once no new highly rated $k_n(\boldsymbol{x})$ are found, the iteration in $m$ terminates and the model with largest $R^2$ is returned. | This work, App. C. |
| CS-$p(\mathcal{M})$ | Comprehensive search Bayesian selection | Of models $\mathcal{M}$ with highest $R^2$ from CS-$R^2$ the model $\mathcal{M}^{(1)}$ with maximal model evidence $p(\boldsymbol{y}|\mathcal{M})$ is returned. | This work, App. C. |
| BSR | Bayesian stepwise regression | Bi-directional stepwise selection using the Bayesian model evidence $p(\boldsymbol{y}|\mathcal{M})$ as score. | (Hastie et al., 2009), this work for $p(\boldsymbol{y}|\mathcal{M})$, App. B. |
| SR3 | Sparse relaxed regression | The regularization is put on an auxiliary variable $u$ and an extra distance term like $\|\boldsymbol{w}-\boldsymbol{u}\|_q$ is added to Eq. (5). | (Zheng et al., 2019). |
| FROLS | Forward regression orthogonal least-squares | Forward stepwise regression selecting terms most correlated with target minimizing Eq. (5) for $q=2$. | (Billings, 2013) |
| STLSQ | Sequentially thresholded least squares | Backward selection stepwise regression using results of regularized regression for $q=2$ in Eq. (5). | (Brunton et al., 2016) |
| MIOSR | Best subset selection via mixed-integer optimized sparse regression | Formulation of regularized regression (5) for $q=0$ as a mixed-integer linear program and specialized algorithms. | (Bertsimas & Gurnee, 2023) |

methods in this experiment. In the second and third experiment the methods are applied to two (chaotic) dynamical systems, where, for the sake of readability, we omitted LARS due to its similar performance compared to LASSO.

In all experiments, we have three features $\boldsymbol{x} = (x_1, x_2, x_3)$, and we used a basis function expansion of polynomials where the power of individual factors was limited to a maximum of $M_1 = 4$, and the combined power of terms $k_n(\boldsymbol{x})$ to a maximum of $M_2 = 6$. For instance, $k(\boldsymbol{x}) = x_1^3 x_2 x_3^2$ would be a valid basis function, while $k(\boldsymbol{x}) = x_1^5 x_2$ would be excluded for exceeding the individual power limit $M_1$, as would $k(\boldsymbol{x}) = x_1^4 x_2 x_3^2$ for exceeding the combined power limit $M_2$. The resulting basis function dimension of $\boldsymbol{K}$ is $p = 72$.

The results of each experiment are illustrated statistically by boxplots, where the box indicates the interquartile range, the whiskers extend by a factor 1.5 beyond the box, and outcomes exceeding the whiskers are shown as individual crosses. The median is shown as a darker horizontal line, the mean as a closed circle.

## 4.1 Random polynomials

For the artificial data, we randomly generated 100 polynomials with 2, 3, and 4 non-zero weights respectively. We restricted the terms of the polynomials to have a maximum collective power $M_2 = 4$, where individual features are restricted to maximum power $M_1 = 2$. The non-zero weights are randomly selected with equal probabilities and their values are uniformly sampled from the set $[-4, -1] \cup [1, 4]$. We generated artificial data $\boldsymbol{X}$ for each polynomial by sampling from normal distributions with means randomly selected from the interval $[-20, 20]$ and standard deviations such that 5% of their probability mass overlap respectively. For the polynomials sized 2, 3, and 4 we generated $N = 20$ and $N = 65$ and $N = 95$ datapoints, respectively. Plugging $\boldsymbol{X}$ into Eq. (1) with $f(\boldsymbol{x}_i)$ given by the random polynomial, and corrupting the output with normal noise with standard deviation $\sigma = 0.01$, we generate data for the response variable $Y$.

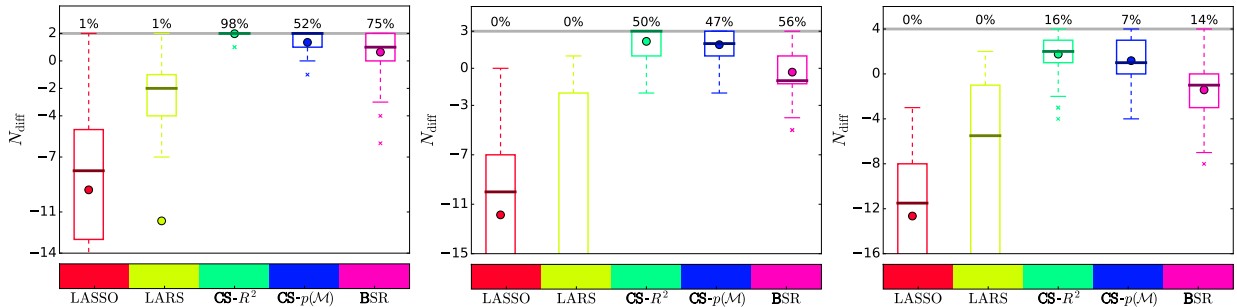

Figure 1: Identification accuracy of learning 100 random polynomials measured in terms of the difference $N_{\text{diff}}$ between the number of true terms found and wrong terms found. We generated polynomials with 2, 3 and 4 terms (left to right), which also constitutes the highest possible value for $N_{\text{diff}}$ and is indicated by a gray horizontal line. The percentages above this line indicate how often all exact terms and no wrong terms have been recovered.

The statistics of the identification accuracy for the 100 polynomials are shown in Fig. 1.

## 4.2 Lorenz system

As a first dynamical system to test our method, we use the chaotic Lorenz system defined as

$$
\begin{aligned}
\dot{x}(t) &= \epsilon\big(y(t) - x(t)\big), \\
\dot{y}(t) &= x\big(\rho - z(t)\big) - y, \\
\dot{z}(t) &= x(t)y(t) - \beta z(t).
\end{aligned}
\tag{9}
$$

The parameters are fixed to its standard values $\epsilon = 10$, $\rho = 28$ and $\beta = 8/3$. As initial condition, we use $(x_0 = -8, y_0 = 8, z_0 = 27)$. We obtain between $N = 200$ and $N = 10000$ data points by solving Eq. (9) numerically for timestep widths between $\Delta t = 0.001$ and $\Delta t = 0.1$. We corrupt the solutions with normal noise levels between $\sigma = 0.001$ and $\sigma = 0.1$. Forcing the simulation time to be larger than $T = 2$, we thus have 80 different scenarios with varying $N$, $\Delta t$, $\sigma$ and $T$. We apply the equation learning methods to all scenarios to obtain some statistics on identification accuracy and mean-absolute error (MAE).

We measure the identification accuracy as the number of correctly identified equations comprising the model (9). The MAE is obtained by solving the learnt dynamical system equations numerically for randomly selected initial values, and comparing the result with the solution we get solving the true model Eq. (9) for the same initial condition.

To further increase the sample size for the statistical evaluation of the methods, and to exclude the possibility to have a particular initial condition that suits a certain method by chance, we sample 100 initial values from the Lorenz attractor and solve each learnt model from all scenarios and equation learning methods for the same 100 initial values.

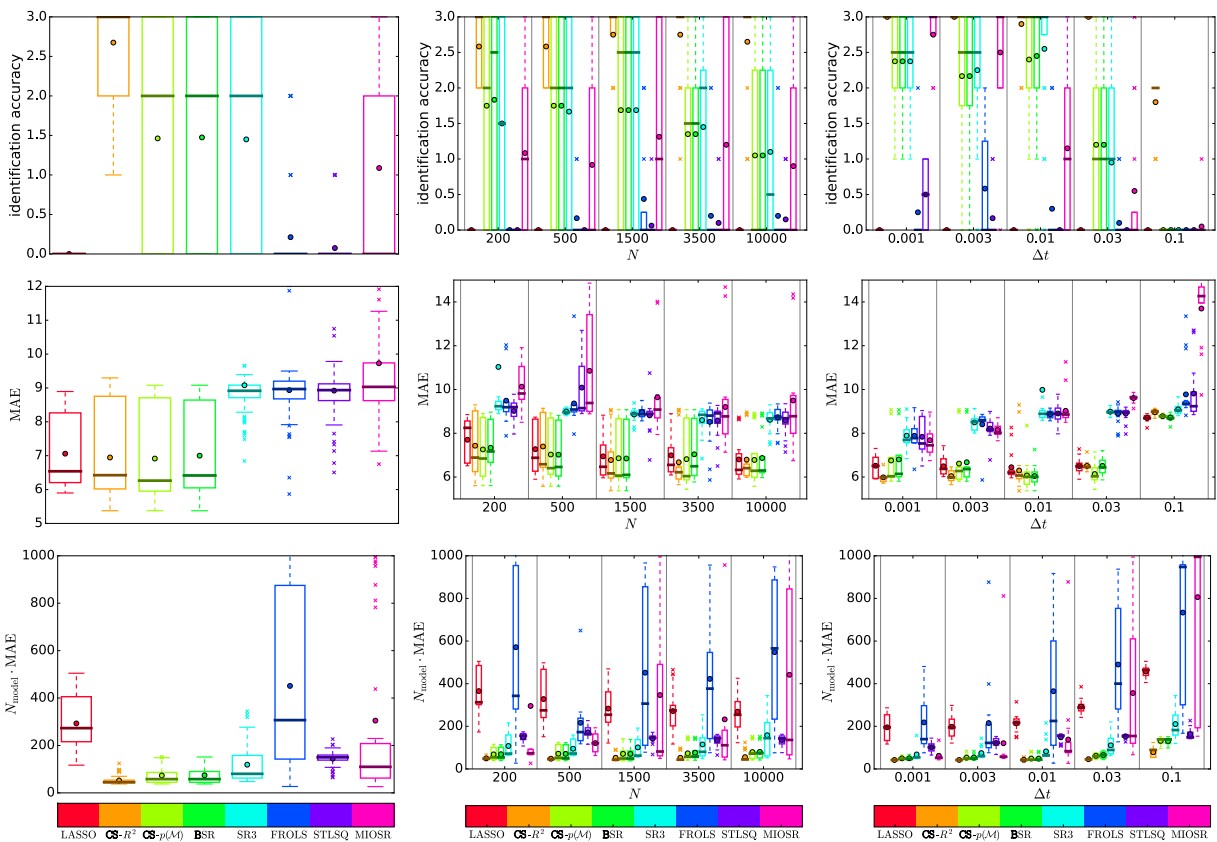

Figure 2: Statistical results for models learnt from data created by solving the Lorenz system (9). In each row of the nine plots a different metric is display: the number of equations identified correctly, the MAE to solutions of the true model, the MAE multiplied by the size of the learnt model. The first column uses the statistics across all scenarios, the second column splits it up in terms of number of datapoints $N$, and the third column in terms of timestep width $\Delta t$.

In Fig. 2 we show the statistical results of our experiment. To emphasize the goal to parsimoniously learn models, we also plot the statistics of the MAE multiplied by the size $N_{model}$ of the learnt model. In addition to the overall statistics, we also show the dependency on $N$ and $\Delta t$ (splitting the data up in terms of $\sigma$ or $T$ did not reveal any insights).

## 4.3 Rabinovich-Fabrikant equations

As a second dynamical system, we use the Rabinovich-Fabrikant equations

$$
\begin{aligned}
\dot{x}(t) &= y(t)\big(z(t) - 1 + x(t)^2\big) + \gamma x(t), \\
\dot{y}(t) &= x(t)\big(3z(t) + 1 - x(t)^2\big) + \gamma y(t), \\
\dot{z}(t) &= -2z(t)\big(\alpha + x(t)y(t)\big).
\end{aligned}
\tag{10}
$$

We choose the parameter values $\alpha = 0.14$ and $\gamma = 0.1$, and the initial conditions $(x_0 = -1.5, y_0 = 0, z_0 = 1)$. We consider 84 scenarios comprising values $N = 1000$ to $N = 16000$, $\Delta t = 0.001$ to $\Delta t = 0.1$, and $\sigma = 0.0001$ and $\sigma = 0.01$, where we enforce $T \geq 5$. We solve Eq. (10) numerically and corrupt the solutions with noise as for the Lorenz system.

The results are shown in Fig. 3 in the same fashion as for the Lorenz system.

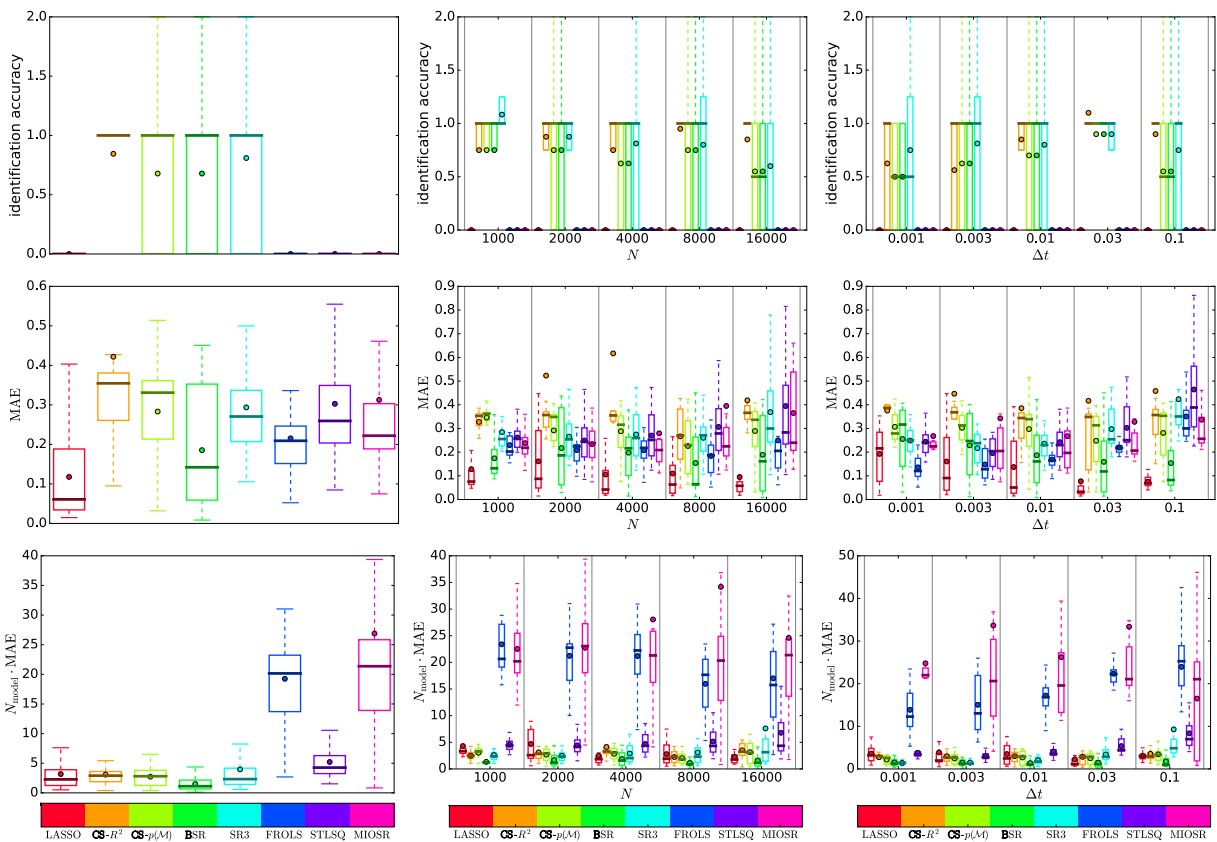

Figure 3: Statistical results from models learnt from data created by solving the Rabinovich-Fabrikant system (10). The plots have the same structure as in Fig. 2: each row shows a different metric, whereas the columns show the overall statistics or split up in terms of $N$ or $\Delta t$.

## 5    Discussion

In this section, we discuss the results of the numerical experiments shown in the previous section.

We begin with the learning of the random polynomials and stress the difficulty of this task, which consists in various aspects: i) Randomly generated polynomials are not hand-picked examples where some fine-tuning is possible. ii) The artificially generated data was not tested to represent the polynomial uniquely. iii) Even if an inferred polynomial deviating from the true polynomial would describe the data well, it does not contribute positively to the identification accuracy.

With these aspects in mind, it is quite remarkable that overall many of the polynomials could be recovered, as shown in Fig. 1. Particularly striking is the success rate of CS-$R^2$ solely based on $R^2$, whereas the LASSO and LARS clearly overfit in terms of model size. However, the overall accuracy clearly declines with increasing number of non-zero terms. This can be explained with a higher chance of terms being selected spuriously leading to too early or too late stopping of exploring the required number of terms. Also the risk of erroneously removal of true terms from $\boldsymbol{K}$ increases. We are confident that more refined stopping and removal criteria can overcome these inaccuracies. In combining several criteria, we see an opportunity to improve the accuracy even further.

The Lorenz system has between 2 and 3 terms per equation, and as such was learnt quite successfully, as the first row of Fig. 2 shows. The highest identification accuracy was again achieved by CS-$R^2$, the LASSO was not able to identify any equation, and of the SINDy methods the relaxed regularized regression SR3 and the best subset-selection using mixed integer optimization MIOSR performed best. In terms of MAE, the

LASSO is among the best, but requires significantly larger models is shown by $N_{\text{model}}$ MAE. Most of the methods are relatively robust against $N$ and $\Delta t$. Interestingly, $N_{\text{model}}$ MAE worsens with more datapoints in the case of MIOSR but improves for the LASSO, signifying that $\ell_0$ regression "sees" more in the data than there is if given enough data, while $\ell_1$ requires more data to produce smaller models (c.f. Eq. (5)). Regarding robustness against larger timesteps, it turns out that MIOSR is particularly sensitive with deteriorating performance for smaller $\Delta t$, as well as FROLS and the LASSO to a lesser extent.

The Rabinovich-Fabrikant system turned out to be particularly difficult to learn. The identification accuracy for the two CS methods, the BSR and the relaxed regularized SR3 was on average just below 1 equation, while the other methods failed to identify any equation in all scenarios. No method was able to learn the complete system of equations correctly. Interestingly, the MAE was still reasonable, in particular the LASSO performed well.

A particular problem in learning dynamical systems is that there is no guarantee that the learnt models are solvable. In cases where the numerical solutions failed, we excluded the results from the statistics and kept a record of how often this happened for the various equation learning methods. For the Lorenz system, LASSO, SR3 and STLSQ failed at about 1%, MIOSR at about 20%, and the other methods always produced solvable systems. This is a little different for the Rabinovich-Fabrikant system, where all methods produced unsolvable models between about 1% and 15%, with STLSQ the most problematic followed by SR3 and CS-$R^2$.

## 6 Conclusions

We extensively tested our methods CS-$R^2$, CS-$p(\mathcal{M})$ and the BSR adaption against the standard methods LASSO and LARS, as well as state-of-the-art methods SR3, FROLS, STLSQ and MIOSR. To our knowledge, we are the first to explore equation learning based on a comprehensive model search outperforming existing state-of-the-art methods in terms of identification accuracy, and at least on equal terms regarding forecast quality.

A disadvantage of our comprehensive search approach is the higher computational cost. However, a direct advantage is that the model evaluation is trivially parallelizable. Also the little amount of data needed for high success rates is striking – tests on two sets of random polynomials where even done with less datapoints than number of basis functions, $N < p$. Testing candidate models individually also allows for great flexibility when it comes to constraints or conditions on models such as solvability, as well as eliminating the risk of getting stuck in local minima of an objective function. It also allows combining several criteria for model and feature selection, in particular complementing existing methods in an independent way with the potential of synergy effects.

The observation that learnt chaotic model comprising all true terms plus a few extra terms can decrease the MAE has an interesting implication worth exploring in a future work: It seems possible to learn correction terms from data that lead to a better forecast horizon than the true model itself.

Finally, we contributed towards the utilization of the Bayesian model evidence $p(\boldsymbol{y}|\mathcal{M})$ in equation learning. Here, we benefit from using a conjugate prior for which $p(\boldsymbol{y}|\mathcal{M})$ can be computed analytically and showed in our numerical experiments that our choice of empirical prior is well suited for the tasks considered. However, in general, one would like to have the freedom to select any prior which may entail particularly computationally costly evidence estimators. Performing the comprehensive search with $R^2$ can boil down the number of candidate models to a feasible number, an approach planned to be explored in a future work.

Considering these possibilities and the promising identification accuracy achieved, we hope to open new avenues of equation learning.

## Acknowledgements

DN gratefully acknowledges support by the African Institute for Mathematical Sciences (AIMS) and the Oppenheimer Memorial Trust (OMP). BB has been supported by the BMBF through the German Research Chair at AIMS administered by the Humboldt Foundation.

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

## A    Linear regression

In this section, we show how linear regression can be applied to equation learning, which sets the basis of this work. Details to regression can be found in Montgomery et al. (2012).

Starting point are $N$ observations $(y_i, x_{ij})$, where the index $i$ denotes data points, and the index $j$ denotes features. We assume that the response (dependent) variable $Y$ is given as a function of explanatory (independent) variable $\boldsymbol{x}$,

$$Y = f(\boldsymbol{x}) + \sigma Z, \tag{11}$$

where $f(\boldsymbol{x})$ defines the model, $Z$ is a standard normal random variable, and $\sigma^2$ is the variance of the noise term. The explanatory observations are often organized in terms of a design matrix $\boldsymbol{X}$, with features in columns and datapoints in rows, $X_{ij} = x_{ij}$.

We assume that $f(\boldsymbol{x})$ can be given in terms of $p$ basis functions $k_n(\boldsymbol{x})$,

$$f(\boldsymbol{x}) = \sum_{n=1}^{p} w_n k_n(\boldsymbol{x}), \tag{12}$$

with weights $w_n$, known as basis function expansion. Similar to the design matrix, we can define a basis function design matrix $\boldsymbol{K}$ with elements $K_{in} = k_n(\boldsymbol{x}_i)$.

The ordinary least squares (OLS) estimates for $w_n$ are known to be

$$\hat{\boldsymbol{w}} = \operatorname*{arg\,min}_{\boldsymbol{w} \in \mathbb{R}^p} \left\| \boldsymbol{K}\boldsymbol{w} - \boldsymbol{y} \right\|_2^2 = (\boldsymbol{K}^{\mathrm{T}} \boldsymbol{K})^{-1} \boldsymbol{K}^{\mathrm{T}} \boldsymbol{y} \tag{13}$$

with the $q$-norm

$$\|\boldsymbol{w}\|_q = \left[ \sum_n |w_n|^q \right]^{1/q}. \tag{14}$$

Predictions based on the OLS estimates are then given by

$$\hat{\boldsymbol{y}} = \boldsymbol{K}\hat{\boldsymbol{w}}. \tag{15}$$

From the normality of linear regression, it is known that the estimates $\hat{\boldsymbol{w}}$ follow a normal distribution with the mean given by the true values for $\boldsymbol{w}$ and the variance given by $\hat{\sigma}^2 = \frac{(\boldsymbol{y} - \hat{\boldsymbol{y}})^{\mathrm{T}}(\boldsymbol{y} - \hat{\boldsymbol{y}})}{N - p}$. We will use these properties later to empirically define the prior in the Bayesian description.

In view of Eq. (13), once a choice of $k_n(\boldsymbol{x})$ is made, the actual learning of the model is straight forward. The difficult part is the choice of $k_n(\boldsymbol{x})$: on the one hand we require sufficient expressivity of the model to minimize bias, on the other hand we want to avoid overfitting to minimize variance of predictions. This bias-variance trade-off essentially dictates the number of $k_n(\boldsymbol{x})$, i.e. the effective dimension of feature space or complexity. Apart from the appropriate model size, we also seek the "correct" $k_n(\boldsymbol{x})$, in the sense that the true $f(\boldsymbol{x})$ is recovered from data generated by Eq. (11).

A common approach to avoid overfitting is regularization,

$$\hat{\boldsymbol{w}} = \operatorname*{arg\,min}_{\boldsymbol{w} \in \mathbb{R}^p} \left[ \left\| \boldsymbol{K}\boldsymbol{w} - \boldsymbol{y} \right\|_2^2 + \lambda \|\boldsymbol{w}\|_q \right], \tag{16}$$

where $\|\boldsymbol{w}\|_q = [\sum_n |w_n|^q]^{1/q}$, and $\lambda$ is the Lagrange parameter that sets the strength of the $\ell_q$ penalty. Common choices for $q$ are $q = 2$ (Ridge regression), $q = 1$ (the standard, sparsity promoting choice known as the LASSO), or combinations such as elastic net. A special case of regularization is $q = 0$, for which $\|\boldsymbol{w}\|_0 = \sum_n \delta_{w_n, 0}$ is the number of non-zero weight estimates – the regression procedure with this penalty is often called *best subset selection* and requires specialized optimization algorithms.

A standard measure for goodness of fit is the coefficient of determination, $R^2$, which relates the variance explained by the prediction $\hat{\boldsymbol{y}}$ to the variance of the observations $\boldsymbol{y}$,

$$R^2 = 1 - \frac{(\boldsymbol{y} - \hat{\boldsymbol{y}})^{\mathrm{T}}(\boldsymbol{y} - \hat{\boldsymbol{y}})}{(\boldsymbol{y} - \bar{y})^{\mathrm{T}}(\boldsymbol{y} - \bar{y})}, \tag{17}$$

where $\bar{y} = \frac{1}{n}\sum_{i=1}^{N} y_i$ is the empirical mean. Assuming standardized $\boldsymbol{y}$, $R^2$ can be simplified to

$$R^2 = 1 - \frac{\boldsymbol{y}^{\mathrm{T}}\boldsymbol{y} - \boldsymbol{y}^{\mathrm{T}}\boldsymbol{K}\hat{\boldsymbol{w}} - \hat{\boldsymbol{w}}^{\mathrm{T}}\boldsymbol{K}^{\mathrm{T}}\boldsymbol{y} + \hat{\boldsymbol{w}}^{\mathrm{T}}\boldsymbol{K}^{\mathrm{T}}\boldsymbol{K}\hat{\boldsymbol{w}}}{(\boldsymbol{y} - \bar{y})^{\mathrm{T}}(\boldsymbol{y} - \bar{y})} \tag{18}$$

$$= 1 - \frac{\boldsymbol{y}^{\mathrm{T}}\boldsymbol{y} - \boldsymbol{y}^{\mathrm{T}}\boldsymbol{K}\hat{\boldsymbol{w}}}{(\boldsymbol{y} - \bar{y})^{\mathrm{T}}(\boldsymbol{y} - \bar{y})} \tag{19}$$

$$= 1 - \frac{\boldsymbol{y}^{\mathrm{T}}\boldsymbol{y} - \hat{\boldsymbol{w}}^{\mathrm{T}}\boldsymbol{K}^{\mathrm{T}}\boldsymbol{y}}{(\boldsymbol{y} - \bar{y})^{\mathrm{T}}(\boldsymbol{y} - \bar{y})} \tag{20}$$

$$= 1 - \frac{\boldsymbol{y}^{\mathrm{T}}\boldsymbol{y} - \hat{\boldsymbol{w}}^{\mathrm{T}}\boldsymbol{K}^{\mathrm{T}}\boldsymbol{y}}{\boldsymbol{y}^{\mathrm{T}}\boldsymbol{y}} \tag{21}$$

$$= \frac{\boldsymbol{y}^{\mathrm{T}}\boldsymbol{K}\left(\boldsymbol{K}^{\mathrm{T}}\boldsymbol{K}\right)^{-1}\boldsymbol{K}^{\mathrm{T}}\boldsymbol{y}}{\boldsymbol{y}^{\mathrm{T}}\boldsymbol{y}}, \tag{22}$$

where we plugged in Eq. (15) in the 1st line, in the 2nd line we used that $\boldsymbol{K}^{\mathrm{T}}\boldsymbol{y} = \boldsymbol{K}^{\mathrm{T}}\boldsymbol{K}\hat{\boldsymbol{w}}$, in the 3rd line that $\boldsymbol{y}^{\mathrm{T}}\boldsymbol{K}\hat{\boldsymbol{w}} = \sum_{jk} y_j K_{jk}\hat{w}_k = \sum_{jk} \hat{w}_k K_{jk} y_j = \hat{\boldsymbol{w}}^{\mathrm{T}}\boldsymbol{K}^{\mathrm{T}}\boldsymbol{y}$, in the 4th line we assumed $\bar{y} = 0$ due to standardization, and in the last line we used $\hat{\boldsymbol{w}} = (\boldsymbol{K}^{\mathrm{T}}\boldsymbol{K})^{-1}\boldsymbol{K}^{\mathrm{T}}\boldsymbol{y}$. For sparse weight estimates $\hat{\boldsymbol{w}}$, $R^2$ is extremely efficient to compute.

A value of $R^2$ close to 1 signifies good predictions $\hat{\boldsymbol{y}}$. However, it is well known that $R^2$ can always be brought closer to 1 by increasing the number of features in the model, thus essentially lacking any overfitting penalty.

For the purpose of adequate model selection, it is helpful to formulate regression in a Bayesian setting. The main step to this end is defining the likelihood distribution for $Y$. In the simple case of Eq. (11), $Y$ is normally distributed,

$$p_{\mathrm{li}}(y|\boldsymbol{w}, \sigma, \mathcal{M}) = \mathcal{N}(\boldsymbol{\mu}, \sigma), \tag{23}$$

where the mean vector is given by $\mu_i = f(\boldsymbol{x}_i)$. Here, we have included the model $\mathcal{M}$ as a condition for the likelihood. In general, the model $\mathcal{M}$ can be specified in any way that determines the likelihood apart from its parameters, e.g. the underlying distribution assumption, but here $\mathcal{M}$ is equivalent to the choice $\boldsymbol{n}$ of terms being part of the model equation $f(\boldsymbol{x})$ in Eq. (12). More specifically, $\boldsymbol{n}$ is a boolean vector of length $p$ matching an ordered list of the terms of the basis function dictionary, indicating with a 1 that the corresponding term is part of the model. The model size is therefore given by $m = \sum_{j=1}^{p} n_j$

Maximization of the corresponding log-likelihood function reproduces the OLS result (13). Encoding existing information on the weights $w_n$ as a prior distribution $p_{\mathrm{pr}}(\boldsymbol{w}, \sigma)$, Bayes' formula implies for the posterior distribution

$$p_{\mathrm{po}}(\boldsymbol{w}, \sigma|\boldsymbol{y}) = \frac{p_{\mathrm{li}}(\boldsymbol{y}|\boldsymbol{w}, \sigma, \mathcal{M})\, p_{\mathrm{pr}}(\boldsymbol{w}, \sigma)}{p(\boldsymbol{y}|\mathcal{M})}, \tag{24}$$

where the normalization factor $p(\boldsymbol{y}|\mathcal{M})$ is known as the *evidence* or prior-predictive value. At this point, we have substituted the $N$-sized data $\boldsymbol{y}$ for the target variable $Y$ under the assumption of a factorizing likelihood distribution.

The marginal likelihood $p(y|\mathcal{M})$ may be interpreted as the likelihood of the observation $y$ given the model $\mathcal{M}$ regardless the choice of parameters. Hence, taking $p(\boldsymbol{y}|\mathcal{M})$ as the likelihood function on the level of models, we may use Bayes' formula again to obtain

$$p_{\mathrm{po}}(\mathcal{M}|\boldsymbol{y}) = \frac{p(\boldsymbol{y}|\mathcal{M})\, p_{\mathrm{pr}}(\mathcal{M})}{p(\boldsymbol{y})}, \tag{25}$$

where $p_{\mathrm{pr}}(\mathcal{M})$ is a prior for a set of models. Without further information on the model set, we may assume a constant $p_{\mathrm{pr}}(\mathcal{M})$, and find that the model evidence $p(\boldsymbol{y}|\mathcal{M})$ is proportional to the probability of a model, $p_{\mathrm{po}}(\mathcal{M}|\boldsymbol{y})$, given the observations $\boldsymbol{y}$. Therefore, if we could maximize $p(\boldsymbol{y}|\mathcal{M})$ over all $\mathcal{M}$, we would in fact identify the model $\hat{\mathcal{M}}$ that most likely explains the observations $y$.

From Eq. (24) it follows that the evidence is given by

$$p(\boldsymbol{y}|\mathcal{M}) = \int \mathrm{d}\sigma \int \mathrm{d}\boldsymbol{w}\, p_{\mathrm{li}}(y|\boldsymbol{w},\sigma,\mathcal{M})\, p_{\mathrm{pr}}(\boldsymbol{w},\sigma), \tag{26}$$

which in general is not solvable analytically, and also poses a particularly tough numerical challenge (Von Der Linden et al., 1999; Knuth et al., 2015). Fortunately, by choosing $p_{\mathrm{pr}}(\boldsymbol{w},\sigma)$ conjugate to $p_{\mathrm{li}}(y|\boldsymbol{w},\sigma,\mathcal{M})$, the integral becomes solvable analytically. The conjugate prior, however, is not necessarily the sensible choice from the inference point of view. In fact, making a good choice for the prior is a much debated problem (Fortuin, 2021). Here, we demonstrate that for the purpose of linear equation learning, the conjugate prior is a suitable choice, if hyper-parameters are distilled from data. The question whether other choices for the prior would perform significantly better is left for future research.

The conjugate prior for the likelihood (23) is the gamma-normal distribution (O'Hagan & Kendall, 1994)

$$p(\boldsymbol{w},\tau|\boldsymbol{\mu},\boldsymbol{\Sigma},k,\vartheta) = \frac{\sqrt{\det \boldsymbol{\Sigma}}}{(2\pi)^{p/2}\Gamma(k)\vartheta^k}\, \tau^{p/2+k-1}\, e^{-\frac{\tau}{2}(\boldsymbol{w}-\boldsymbol{\mu})^{\mathrm{T}}\boldsymbol{\Sigma}(\boldsymbol{w}-\boldsymbol{\mu})-\tau/\vartheta} \tag{27}$$

with mean vector $\boldsymbol{\mu}$ and precision matrix $\boldsymbol{\Sigma}$ for the weights $\boldsymbol{w}$, and shape $k$ and scale $\vartheta$ for the precision $\tau = 1/\sigma^2$. Plugging Eq. (27) and Eq. (23) into Eq. (26) and performing the integration, we obtain for the log-evidence per data-point the closed expression

$$\frac{1}{N}\ln p(\boldsymbol{y}|\mathcal{M}) = \frac{1}{2N}\ln\frac{\det \boldsymbol{\Sigma}}{\det \boldsymbol{A}} - \frac{1}{2}\ln 2\pi - \left(\frac{1}{2}+\frac{k}{N}\right)\ln\left(\frac{\xi}{2}+\frac{1}{\vartheta}\right)$$
$$- \frac{k}{N}\ln\vartheta + \frac{1}{N}\ln\Gamma\left(\frac{N}{2}+k\right) - \frac{1}{N}\ln\Gamma(k) \tag{28}$$

with

$$\boldsymbol{A} = \boldsymbol{K}^{\mathrm{T}}\boldsymbol{K} + \boldsymbol{\Sigma}, \tag{29}$$

$$\boldsymbol{b} = \boldsymbol{K}^{\mathrm{T}}\boldsymbol{y} + \boldsymbol{\Sigma}\boldsymbol{\mu}, \tag{30}$$

$$\xi = \boldsymbol{y}^{\mathrm{T}}\boldsymbol{y} + \boldsymbol{\mu}^{\mathrm{T}}\boldsymbol{\Sigma}\boldsymbol{\mu} - \boldsymbol{b}^{\mathrm{T}}\boldsymbol{A}^{-1}\boldsymbol{b}. \tag{31}$$

We detail the calculations in App. B.

Owing to the normality of linear regression, and from standardizing the data, it is reasonable to assume the following parameters for the prior: For the mean vector, we choose $\boldsymbol{\mu} = \hat{\boldsymbol{w}}$, and the precision matrix $\boldsymbol{\Sigma}$ is taken to be diagonal with elements $\mathrm{diag}(\boldsymbol{\Sigma}) = \frac{1-p/N}{\boldsymbol{y}^{\mathrm{T}}\boldsymbol{y}-\hat{\boldsymbol{w}}^{\mathrm{T}}\boldsymbol{K}^{\mathrm{T}}\boldsymbol{y}}$, resulting in normal distributions broadened by a factor $N$ to make the prior more uninformative. The gamma distribution entering Eq. (27) has the mode $(k-1)\vartheta$, which we set to 1 due to standardized $y$. The scale is set to $\vartheta = 1/2$ which appears to be broad enough for an uninformative prior.

For completeness, we mention a few more selection criteria. The adjusted $R^2$ (Montgomery et al., 2012)

$$R_{\mathrm{adj}}^2 = 1 - \frac{N-1}{N-m-1}\,(1-R^2) \tag{32}$$

equips the usual $R^2$ with an overfitting penalty. The Akaike information criterion (AIC) measures the loss of information by using the inferred model instead of the (unknown) true model (Murphy, 2012), and similarly but derived from the model evidence (26) in the big data limit, follows the Bayesian (Schwarz) information criterion,

$$\mathrm{AIC} = 2p_{\mathrm{li}}(\boldsymbol{y}|\hat{\boldsymbol{w}},\hat{\sigma}) - 2m\,, \quad \mathrm{BIC} = p_{\mathrm{li}}(\boldsymbol{y}|\hat{\boldsymbol{w}},\hat{\sigma}) - 2m\ln N \tag{33}$$

Similarly, but derived from the model evidence (34) in the big data limit, is the Bayesian (Schwarz) information criterion,

$$\text{BIC} = p_{\text{li}}(\boldsymbol{y}|\hat{\boldsymbol{w}}, \hat{\sigma}) - 2m \ln N. \tag{34}$$

Apart from the equations that can directly be written in the form of Eq. (12), a prominent application of linear equation learning is the sparse identification of dynamical systems,

$$\dot{\boldsymbol{x}}(t) = f(\boldsymbol{x}(t)), \tag{35}$$

where $\dot{x}(t)$ denotes the time derivative of $x(t)$. To map this problem to Eq. (12), the response variable can be computed from finite differences $y_i = \frac{x_{i+1} - x_i}{\Delta t}$ for a fixed time step $\Delta t$. In a different approach, the weak form of the differential equations can be utilized; we refer the reader to Messenger & Bortz (2021a) for details.

A restriction for the regression models to stay linear in its parameters is that the parameters of basis functions only enter as weights $w$. Basis functions like

$$e^{ax}, \; \ln(a + x), \; \cos ax, \; x^a, \; \frac{1}{(a + x)^m}, \; \ldots \tag{36}$$

with internal parameter $a$ are not suitable.

This restriction might seem quite limiting. On the other hand, the function $f(\boldsymbol{x}(t), \boldsymbol{w})$ defining a dynamical system typically is linear in its parameters $\boldsymbol{w}$. The reason for that is that functions shown in Eq. (36) often reproduce when differentiated, which can be used to eliminate these functions, retrieving the standard form shown in Eq. (12) and Eq. (11). being linear in parameters. Some special functions like Bessel, Hankel, Struve and Meijer functions are even defined as solutions of differential equations.

In general, by considering the differentiated response variable,

$$y \;\mapsto\; \frac{\mathrm{d}y}{\mathrm{d}x} \simeq \frac{y(x_{i+1}) - y(x_i)}{x_{i+1} - x_i}, \tag{37}$$

if necessary to higher order, we can learn a surprisingly broad class of equations relating $y$ and $\boldsymbol{x}$, even relations that do not exist in closed form. Here, we restrict ourselves to dynamical systems and leave the full exploration of learning in this broad model class for future work.

While many equations with non-linear parameters can be rewritten in linear form by differentiation as explained above, some functions like $x^a$, $\frac{1}{(a+x)^m}$ can only be reduced to fractions or require many differentiations which can give rise to numerical issues. Therefore, if we were able to include fractions in the basis function expansion, we could expand the model class even further. The problem is that basis functions like $\frac{1}{1+x^n}$ are divergent for certain values of $x$ and odd $n$.

It is, however, possible to modify the regression model to also incorporate fractions. In its simplest form, we may consider

$$y = \frac{\sum_{n=1}^{p} w_n \, k_n(\boldsymbol{x}) + \sigma z}{\sum_{m=1}^{p} v_m \, k_m(\boldsymbol{x})} \tag{38}$$

with different (sparse) weights $v_m$ but same set of basis functions for the denominator. For the next step, we assume that $k_n(\boldsymbol{x})$ is part of the numerator, that is $w_1 \neq 0$, and we can rewrite

$$k_1(\boldsymbol{x}) = \sum_{m=1}^{p} \frac{v_m}{w_1} \, y k_m(\boldsymbol{x}) - \sum_{n=2}^{p} \frac{w_n}{w_1} \, k_n(\boldsymbol{x}). \tag{39}$$

In this form, $k_1$ takes the role of the response variable, and we have a second set of basis functions given by $y k_m$. For a given model of this form, the weights also follow deterministically from Eq. (13), only $w_1$ needs to be determined from a 1-dimensional numerical root-finding algorithm. We leave this ansatz for future research.

A similar idea has been proposed in Kaheman et al. (2020), where a $\ell_0$ regularized objective function needs to be minimized for each possible basis function taking the role of $k_1$ above. Our comprehensive search strategy naturally includes this procedure as a straight forward possibility, which is planned to be investigated in future work.

## B  Exact Bayesian model evidence

The model is given by

$$y_i = \sum_n w_n K_{in} + z_i/\sqrt{\tau} \tag{40}$$

where $K_{in} = k_n(\boldsymbol{x}_i)$ is the basis function design matrix, $w_n$ are the weights, $\tau = 1/\sigma^2$ is the precision, and $z_i \sim \mathcal{N}(0, 1)$. For the whole vector $\boldsymbol{y}$ of $N$ responses, we can use the multivariate normal for the likelihood,

$$p(\boldsymbol{y} \,|\, \boldsymbol{K}, \boldsymbol{w}, \tau) = \frac{\tau^{N/2}}{(2\pi)^{N/2}} \exp\Big( -\frac{\tau}{2}\,(\boldsymbol{y} - \boldsymbol{K}\boldsymbol{w})^{\mathrm{T}}\,(\boldsymbol{y} - \boldsymbol{K}\boldsymbol{w}) \Big), \tag{41}$$

where the precision matrix is diagonal with identical $\tau$ on the diagonal. Since the weight parameters $w_n$ enter quadratically, we can rewrite this expression in normal form for $\boldsymbol{w}$,

$$
\begin{aligned}
p(\boldsymbol{y}|K, \boldsymbol{w}, \tau) = {} & \frac{\tau^{N/2}}{(2\pi)^{N/2}} \exp\Big( -\frac{\tau}{2}\,S \Big) \\
& \times \exp\Big( -\frac{\tau}{2}\,(\boldsymbol{w} - \hat{\boldsymbol{w}})^{\mathrm{T}}\,\boldsymbol{K}^{\mathrm{T}}\boldsymbol{K}\,(\boldsymbol{w} - \hat{\boldsymbol{w}}) \Big)
\end{aligned}
\tag{42}
$$

with the residual sum of squares

$$
\begin{aligned}
S &= (\boldsymbol{y} - \boldsymbol{K}\hat{\boldsymbol{w}})^{\mathrm{T}}(\boldsymbol{y} - \boldsymbol{K}\hat{\boldsymbol{w}}) && \tag{43} \\
&= \boldsymbol{y}^{\mathrm{T}}\boldsymbol{y} - \hat{\boldsymbol{w}}^{\mathrm{T}}\boldsymbol{K}^{\mathrm{T}}\boldsymbol{y} && \tag{44} \\
&= \boldsymbol{y}^{\mathrm{T}}\boldsymbol{y} - \boldsymbol{y}^{\mathrm{T}}\boldsymbol{K}(\boldsymbol{K}^{\mathrm{T}}\boldsymbol{K})^{-1}\boldsymbol{K}^{\mathrm{T}}\boldsymbol{y} && \tag{45}
\end{aligned}
$$

The mixed terms cancel after plugging in $\boldsymbol{K}^{\mathrm{T}}\boldsymbol{y} = \boldsymbol{K}^{\mathrm{T}}\boldsymbol{K}\hat{\boldsymbol{w}}$ from the known OLS solution $\hat{\boldsymbol{w}} = (\boldsymbol{K}^{\mathrm{T}}\boldsymbol{K})^{-1}\boldsymbol{K}^{\mathrm{T}}\boldsymbol{y}$.

The above is of the form of a gamma distribution for $\tau$ multiplied with a normal distribution for $\boldsymbol{w}$ conditioned on $\tau$. If we use a prior of the same form, we keep the form for the posterior, and thus have found the conjugate prior.

As a prior for the weights $\boldsymbol{w}$, we choose

$$p(\boldsymbol{w} \,|\, \boldsymbol{\mu}, \boldsymbol{\Sigma}) = \frac{\tau^{p/2}\sqrt{\det\boldsymbol{\Sigma}}}{(2\pi)^{p/2}} \exp\Big( -\frac{\tau}{2}\,(\boldsymbol{w} - \boldsymbol{\mu})^{\mathrm{T}}\,\boldsymbol{\Sigma}\,(\boldsymbol{w} - \boldsymbol{\mu}) \Big), \tag{46}$$

where $\tau\boldsymbol{\Sigma}$ is the precision matrix with $\tau$ split off, and $\boldsymbol{\mu}$ is the mean vector of the multivariate normal prior. Splitting off $\tau$ technical means that specifying $\boldsymbol{\Sigma}$ is relative to the unknown $\tau$, but $\tau$ does not need to be known for that, as we integrate over all possible $\tau$ values.

For the posterior, we are interested in the quadratic form involving $\boldsymbol{w}$,

$$
\begin{aligned}
& (\boldsymbol{w} - \hat{\boldsymbol{w}})^{\mathrm{T}}\,\boldsymbol{K}^{\mathrm{T}}\boldsymbol{K}\,(\boldsymbol{w} - \hat{\boldsymbol{w}}) + (\boldsymbol{w} - \boldsymbol{\mu})^{\mathrm{T}}\,\boldsymbol{\Sigma}\,(\boldsymbol{w} - \boldsymbol{\mu}) && \tag{47} \\
& = \boldsymbol{w}^{\mathrm{T}}\,\boldsymbol{A}\,\boldsymbol{w} - 2\,\boldsymbol{w}^{\mathrm{T}}\,\boldsymbol{b} + c && \tag{48}
\end{aligned}
$$

with

$$\boldsymbol{A} = \boldsymbol{K}^{\mathrm{T}}\boldsymbol{K} + \boldsymbol{\Sigma}, \tag{49}$$

$$\boldsymbol{b} = \boldsymbol{K}^{\mathrm{T}}\boldsymbol{K}\hat{\boldsymbol{w}} + \boldsymbol{\Sigma}\boldsymbol{\mu}$$
$$= \boldsymbol{K}^{\mathrm{T}}\boldsymbol{y} + \boldsymbol{\Sigma}\boldsymbol{\mu}, \tag{50}$$

$$c = \hat{\boldsymbol{w}}^{\mathrm{T}}\boldsymbol{K}^{\mathrm{T}}\boldsymbol{K}\hat{\boldsymbol{w}} + \boldsymbol{\mu}^{\mathrm{T}}\boldsymbol{\Sigma}\boldsymbol{\mu}$$
$$= \hat{\boldsymbol{w}}^{\mathrm{T}}\boldsymbol{K}^{\mathrm{T}}\boldsymbol{y} + \boldsymbol{\mu}^{\mathrm{T}}\boldsymbol{\Sigma}\boldsymbol{\mu} \tag{51}$$
$$= \boldsymbol{y}^{\mathrm{T}}\boldsymbol{K}(\boldsymbol{K}^{\mathrm{T}}\boldsymbol{K})^{-1}\boldsymbol{K}^{\mathrm{T}}\boldsymbol{y} + \boldsymbol{\mu}^{\mathrm{T}}\boldsymbol{\Sigma}\boldsymbol{\mu}. \tag{52}$$

Put into this form, we can use the Gaussian integral $\int \mathrm{d}^n\boldsymbol{x}\, e^{-\frac{1}{2}\boldsymbol{x}^{\mathrm{T}}\boldsymbol{A}\boldsymbol{x}+\boldsymbol{b}^{\mathrm{T}}\boldsymbol{x}} = \sqrt{\frac{(2\pi)^n}{\det \boldsymbol{A}}}\, e^{\frac{1}{2}\boldsymbol{b}^{\mathrm{T}}\boldsymbol{A}^{-1}\boldsymbol{b}}$ to marginalize,

$$p(\boldsymbol{y}\,|\,\tau) = p(\boldsymbol{y}\,|\,\boldsymbol{K},\tau,\boldsymbol{\mu},\boldsymbol{\Sigma}) \tag{53}$$

$$= \int \mathrm{d}\boldsymbol{w}\, p(\boldsymbol{y}\,|\,\boldsymbol{K},\boldsymbol{w},\tau)\, p(\boldsymbol{w}\,|\,\boldsymbol{\mu},\boldsymbol{\Sigma}) \tag{54}$$

$$= \frac{\sqrt{\tau^{N+p}\det \boldsymbol{\Sigma}}}{\sqrt{(2\pi)^{N+p}}}\, e^{-\frac{\tau}{2}(S+c)} \int \mathrm{d}\boldsymbol{w}\, e^{-\frac{\tau}{2}(\boldsymbol{w}^{\mathrm{T}}\boldsymbol{A}\boldsymbol{w}-2\boldsymbol{b}^{\mathrm{T}}\boldsymbol{w})} \tag{55}$$

$$= \frac{\sqrt{\tau^{N+p}\det \boldsymbol{\Sigma}}}{\sqrt{(2\pi)^{N+p}}}\, e^{-\frac{\tau}{2}(S+c)}\, \frac{\sqrt{(2\pi)^p}}{\sqrt{\tau^p \det \boldsymbol{A}}}\, e^{\frac{\tau}{2}\boldsymbol{b}^{\mathrm{T}}\boldsymbol{A}^{-1}\boldsymbol{b}} \tag{56}$$

$$= \sqrt{\frac{\tau^N \det \boldsymbol{\Sigma}}{(2\pi)^N \det \boldsymbol{A}}}\, e^{-\frac{\tau}{2}(\boldsymbol{y}^{\mathrm{T}}\boldsymbol{y}+\boldsymbol{\mu}^{\mathrm{T}}\boldsymbol{\Sigma}\boldsymbol{\mu}-\boldsymbol{b}^{\mathrm{T}}\boldsymbol{A}^{-1}\boldsymbol{b})}. \tag{57}$$

For the $\tau$-integration, we choose the gamma distribution

$$p(\tau|k,\vartheta) = \frac{1}{\Gamma(k)\vartheta^k}\, \tau^{k-1}\, \exp\big(-\tau/\vartheta\big) \tag{58}$$

as the (conjugate) prior for $\tau$, and define

$$\xi = \boldsymbol{y}^{\mathrm{T}}\boldsymbol{y} + \boldsymbol{\mu}^{\mathrm{T}}\boldsymbol{\Sigma}\boldsymbol{\mu} - \boldsymbol{b}^{\mathrm{T}}\boldsymbol{A}^{-1}\boldsymbol{b}. \tag{59}$$

The remaining $\tau$-integral follows then from $\int_0^\infty \mathrm{d}\tau\, \tau^{c_0} e^{-c_1\tau} = c_1^{-c_0-1}\,\Gamma(c_0+1)$ as

$$p(\boldsymbol{y}) = p(\boldsymbol{y}\,|\,\boldsymbol{K},\boldsymbol{\mu},\boldsymbol{\Sigma},k,\vartheta) \tag{60}$$

$$= \int_0^\infty \mathrm{d}\tau\, p(\tau|k,\vartheta)\, p(\boldsymbol{y}\,|\,\tau) \tag{61}$$

$$= \frac{1}{\Gamma(k)\vartheta^k(2\pi)^{\frac{N}{2}}} \sqrt{\frac{\det \boldsymbol{\Sigma}}{\det \boldsymbol{A}}} \int_0^\infty \mathrm{d}\tau\, \tau^{\frac{N}{2}+k-1}\, e^{-\tau(\frac{\xi}{2}+\frac{1}{\vartheta})} \tag{62}$$

$$= \frac{\Gamma(\frac{N}{2}+k)}{\Gamma(k)\vartheta^k(2\pi)^{\frac{N}{2}}} \sqrt{\frac{\det \boldsymbol{\Sigma}}{\det \boldsymbol{A}}} \left(\frac{\xi}{2}+\frac{1}{\vartheta}\right)^{-\frac{N}{2}-k} \tag{63}$$

and for the log-evidence per data-point we obtain

$$\frac{1}{N}\ln p(\boldsymbol{y}) = \frac{1}{2N}\ln \frac{\det \boldsymbol{\Sigma}}{\det \boldsymbol{A}} - \frac{1}{2}\ln 2\pi$$
$$- \left(\frac{1}{2}+\frac{k}{N}\right)\ln\left(\frac{\xi}{2}+\frac{1}{\vartheta}\right) - \frac{k}{N}\ln \vartheta$$
$$+ \frac{1}{N}\ln\Gamma\left(\frac{N}{2}+k\right) - \frac{1}{N}\ln\Gamma(k). \tag{64}$$

## C  Details and illustration of comprehensive search equation learning

To our knowledge, all equation learning approaches that consider the full candidate model space include an optimization step in various representation spaces of equations. Here, we propose a strategy that does without any numerical optimization algorithms, and instead considers candidate models individually in an almost exhaustive manner. Since already small dictionaries of basis functions can lead to tremendous numbers of candidate models, a combination of cheap selection criteria and successive reduction of model space with a suitable stopping criterion is required. We demonstrate how the simple criterion $R^2$ can be used for such a comprehensive search.

In a first step, a dictionary of basis functions is generated using Alg. 1. These basis functions consist of all possible products of available features $x_j$. In these products, the factors are raised to all possible combinations of powers (line 9), where we restrict individual powers to $M_1$ (line 4) and the combined power of a term to $M_2$ (line 7). For example, for $M_1 = 3$ and $M_2 = 5$, the term $x_1^4 x_3$ would not be allowed since $4 > M_1$, and the term $x_1^2 x_2 x_3^3$ would not be allowed because $2 + 1 + 3 > M_2$.

The comprehensive search (CS) strategy we propose is based on this basis function expansion and described by Alg. 3. It begins by considering all regression models with $m$ non-zero weights $\boldsymbol{w}_j$ (line 9), c.f. Eq. (12). To this end, the auxiliary Alg. 2 produces a list of models with top $R^2$ values by looping through all candidate models of size $m$. These models are returned as an index matrix $\boldsymbol{M}$, indicating selected terms with a 1 and deselected terms with a 0, where each column stands for a candidate model (lines 5,11). The models are sorted in descending order with respect to $R^2$ (line 13).

Back to Alg. 3, we successively increase $m$ starting from $m = 1$ (lines 7-8). We found that for a fixed $m$ value, $R^2$ performs particularly well in identifying the best model out of the millions of models (see for instance the left plot in figure 4). To infer a value for $m$ with just $R^2$, we create a feature rating matrix $\boldsymbol{F}$ defined as the weighted counts of terms being selected across $s$ top models, where the weight is given by $R^2$. Based on $\boldsymbol{F}$, we check for terms selected by $R^2$ for two successive model sizes $m$ and $m-1$ (lines 16-21). If for both $m$ and $m-1$ the same terms are selected consistently, we choose these two terms as part of the inferred model $\mathcal{M}^*$ by CS-$R^2$ and conclude the search.

As for larger values of $m$ the number of candidate models can easily reach hundreds of millions, we implement another strategy to divide out the list of candidate models. If terms have not been selected for two successive model sizes $m$ and $m-1$, we remove these terms from the design matrix $\boldsymbol{K}$ (lines 14-15). In this way, we reduce the model space as we go along.

The rationale for this selection and elimination strategy being solely based on $R^2$ is the following: If the true model has $m^*$ terms, and we are testing all models with $m = m^* - 2$ terms, then the models with largest $R^2$ will consistently be composed of the $m^* - 2$ terms that contribute the most to explaining the variance of $y$. The other 2 true terms will be selected sporadically but at least once, terms that have not been selected at all can hence be removed from the candidate models. Testing in the next stage all models with $m = m^* - 1$ terms, one more term will be consistently selected. The same holds for testing models with $m^*$ terms, but when testing models with $m^* + 1$ terms, no new term can contribute consistently to explaining more of the variance of $y$. While the $R^2$ measure will increase for models with $m^* + 1$ terms, compared to models with $m^*$ terms, no extra term will consistently be selected. Therefore, once no new term is selected consistently when incrementing the number $m$ of terms, we may conclude that all contributing terms have been found. An illustration of this strategy can be found in Fig. 4, where a typical case with $m^* = 3$ is shown.

In a final step, the list of top models from the $R^2$ evaluation can be combined and each tested with the Bayesian model evidence $p(\boldsymbol{y}|\mathcal{M})$ (c.f. App. B). In Fig. 5 the selective power of $p(\boldsymbol{y}|\mathcal{M})$ is demonstrated. Fig. 6 schematically summarizes the CS approach.

The hyperparameters of our procedure are the number $s$ of models used to count consistent selection of basis functions $k_n(\boldsymbol{x})$, together with the threshold $c_{\min}$ that (normalized) count of a feature must exceed to be selected, the number $t$ of the best models in terms of $R^2$ that are combined in a new list of top models for re-evaluation with $p(\boldsymbol{y}|\mathcal{M})$, and the maximum number of iterations, $m_{\max}$. We found that the universally best values are $s = p/2$, where $p$ is the number of basis functions, $c_{\min} = 0.75$, $t = 25$, and $m_{\max} = 8$.

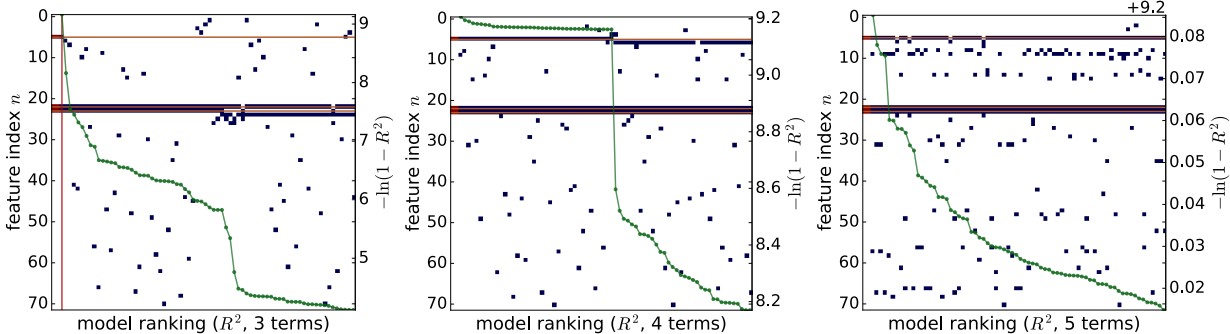

Figure 4: An example of how candidate models with $m=3$, $m=4$, and $m=5$ terms with largest $R^2$ in each category tend to consistently choose the true terms of the model. The indices of all terms in the dictionary (the basis functions $k_n(\boldsymbol{x})$) are shown on the left vertical axis. Each candidate model along the horizontal axis is represented by squares indicating which terms make up the respective model. The ground truth model in this example has 3 terms, indicated by two lighter squares on the left and the horizontal lines. The case where the candidate model is the true model is indicated by a vertical line in the middle plot. The $R^2$ value in a logarithmic scale is shown as a line with closed circles, the values are given by the right vertical axis. It can be seen that the true model is chosen by $R^2$ among all models with 3 terms.

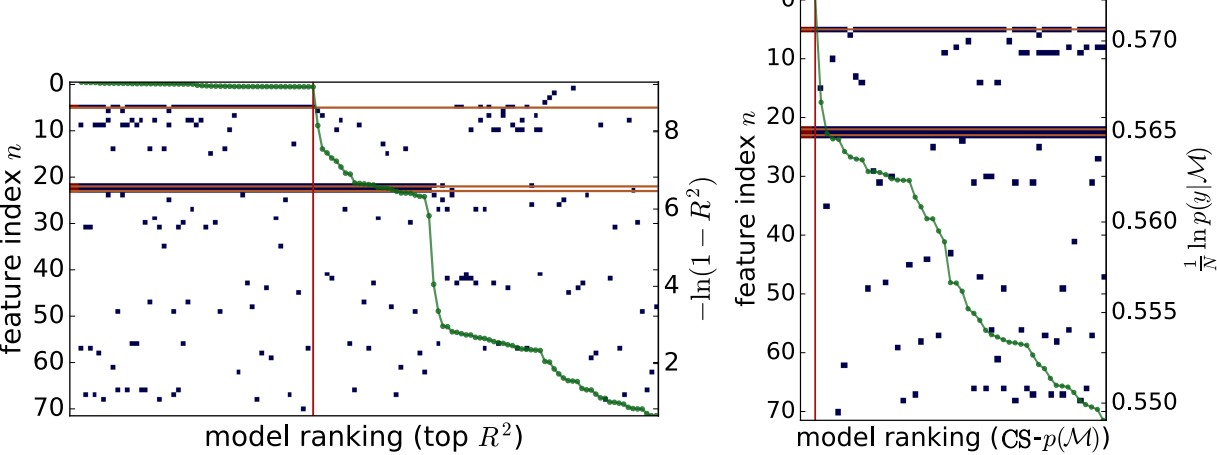

Figure 5: In the plot on the left, the $t$ best models in terms of $R^2$ from each of the $m$-sized candidate models in figure 4 have been combined and sorting according to their $R^2$ value. Clearly, the models with more terms are favored over models with fewer terms, and the true model is not selected. The plot on the rights shows the best models now in terms of the Bayesian model evidence $p(\boldsymbol{y}|\mathcal{M})$, where now the true model is selected illustrating the adequate overfitting penalty held by $p(\boldsymbol{y}|\mathcal{M})$.

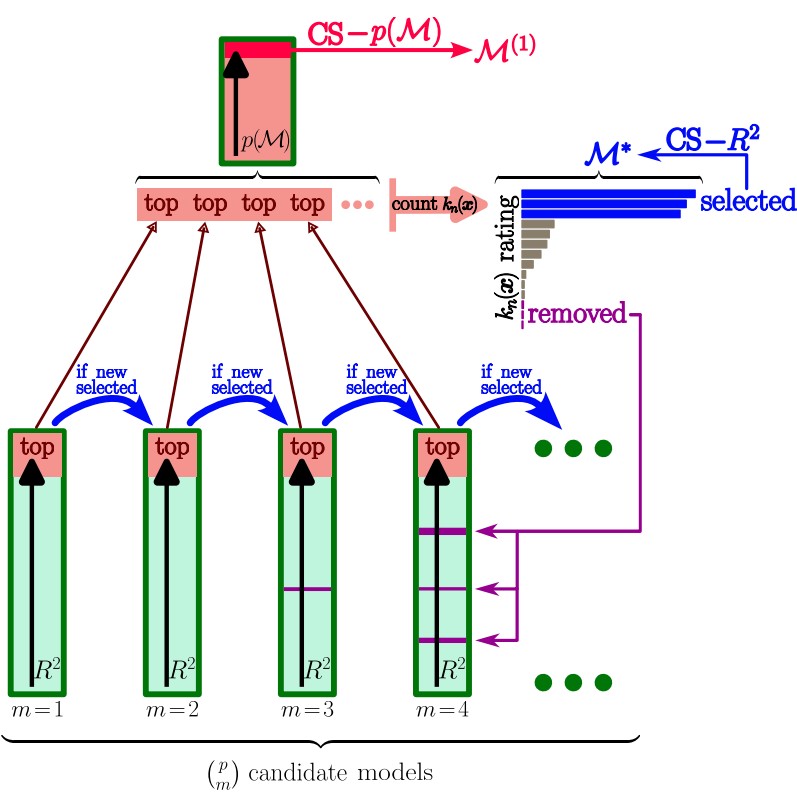

Figure 6: Schematic representation of algorithm CS-$R^2$ and CS-$p\mathcal{M}$. Starting point is the set of candidate models built from $p$ basis functions $k_n(\boldsymbol{x})$. For fixed model size $m$, $R^2$ is computed for all $\binom{p}{m}$ candidate models. Incrementing $m$, top models in terms of $R^2$ are collected, from which basis functions $k_n(\boldsymbol{x})$ are rated based on counts of $k_n(\boldsymbol{x})$ in these top models. Typically, incrementing $m$, new $k_n(\boldsymbol{s})$ with high rates are found and selected for the inferred model $\mathcal{M}^*$ by algorithm CS-$R^2$ as long as $m$ is smaller than the true model size. The iteration in $m$ therefore terminates if no new $k_n(\boldsymbol{s})$ are selected and $\mathcal{M}^*$ is returned. Basis functions $k_n(\boldsymbol{x}$ that are hardly selected are removed and not considered for building candidate models in following iterations. The top models selected by $R^2$ are scored again by $p(\mathcal{M})$ of which the best model $\mathcal{M}^{(1)}$ is the output of algorithm CS-$p(\mathcal{M})$.

The final aspect to be discussed here is the complexity of your CS approach. As an empirical method with unpredictable stopping criterion and pruning, a rigorous analysis appears intractable. However, the complexity of a direct computation of $R^2(\boldsymbol{K}) \sim \boldsymbol{y}^\mathrm{T} \boldsymbol{K} \, (\boldsymbol{K}^\mathrm{T} \boldsymbol{K})^{-1} \, \boldsymbol{K}^\mathrm{T} \boldsymbol{y}$ with $N \times m$ matrix $\boldsymbol{K}$ and $N \times 1$ vector $\boldsymbol{y}$, c.f. Eq. (6), can be determined as follows: The matrix product $\boldsymbol{K}^\mathrm{T} \boldsymbol{K}$ has complexity $\mathcal{O}(m^2 N)$, matrix-vector products $\boldsymbol{y}^\mathrm{T} \boldsymbol{K}$ and $\boldsymbol{K}^\mathrm{T} \boldsymbol{y}$ have complexity $\mathcal{O}(mN)$, the matrix-inverse of $\boldsymbol{K}^\mathrm{T} \boldsymbol{K}$ has complexity $\mathcal{O}(m^3)$, such that in total we have a complexity of $\mathcal{O}\big(m^2(m + N)\big)$. Luckily, as discussed in RUser4512 (2018), efficient numerical implementations can bring it down to $\mathcal{O}(m^{1.3} N^{0.7})$, i.e. a near-linear scaling, which we confirmed with an own numerical analysis shown in Fig. 7.

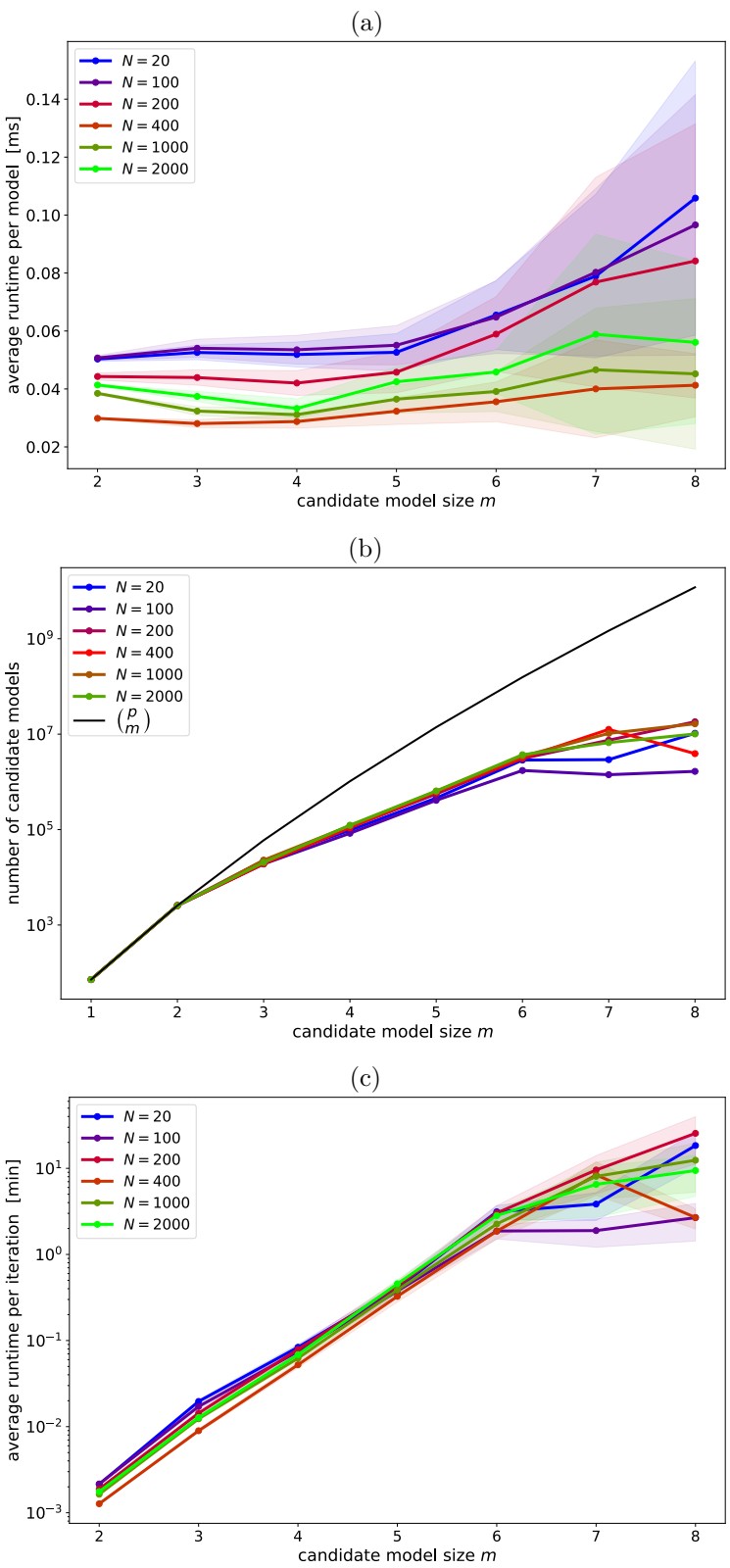

Figure 7: Runtime analysis of the $R^2$ elimination in the comprehensive search approach based on 100 random polynomials and $p = 72$ basis functions on a standard laptop using about 4 cores. Chart (a) shows that the computation of $R^2$ per model stays well below $1\,\text{ms}$ with slight increase with model size $m$. Despite the near-exponential increase of model space dimension shown in chart (b), the efficient computation of $R^2$ and the effect of pruning keeps the runtime per iteration below an exponential increase and for practically relevant model sizes of $m < 5$ well below a minute.

