# OpenReview forum: "Improved identification accuracy in equation learning via comprehensive $\boldsymbol{R^2}$-elimination and Bayesian model selection"
_TMLR — Accepted by TMLR_

### Review · Reviewer_mC8n · 2023-07-02

**Summary Of Contributions:**

The paper proposes a new method for sparse linear regression based on Bayesian model selection, inspired by stepwise regression. Essentially, the algorithm does some less greedy operation when selecting features since it focuses on a larger search space. The algorithm is purely heuristic based and the authors provided some experiments that shows they are better than all existing ones.

**Audience:**

No

**Broader Impact Concerns:**

No specific broader impact concerns.

**Claims And Evidence:**

No

**Requested Changes:**

I would suggest the following:
* Write the algorithm more formally (even it is heuristic, it should be written in a semi-pseudo algorithm format).
* Analyze the computation complexity further.
* Explain more on the comments related to existing literature: e.g., "Greedy algorithms have proven to be more successful than LASSO in equation learning" -- where is the proof? and is it suggesting that SR is always better than LASSO for linear regression?

**Strengths And Weaknesses:**

Strength:
* The idea of the paper is straightforward: a heuristic algorithm is proposed based on comparing existing methods, and experiments are conducted to compare the new method with previous ones.

Weakness:
* The writing can be improved.
* If you are comparing with LASSO and SR, I don't understand why there is a need to discuss those basis functions instead of just focus on linear regression.
* The R2 related discussion seems quite obvious and relatively redundant.
* "Greedy" v.s. "Less greedy" doesn't seem like an intriguing conceptual comparison to me.

---

> ### Author Response · Authors · 2023-07-04
> **Response to Reviewer**
>
> We thank the reviewer for their time and their constructive feedback. We briefly respond to the points raised.
>
> The summary states our algorithm is purely heuristic based. While we agree that our approach is heuristic, we still wish to stress that we have a theoretical motivation (section 3.4) for our approach, just like, for instance, regularization to regression was theoretically motivated but not theoretically guaranteed to work in general.
>
> Our response to the 4 stated weaknesses:
>
> (1) We will take care to improve the writing.
>
> (2) LASSO is known to work well for reproduction in compressed sensing if the covariates are only weakly correlated. If correlations are present, LASSO can still minimize variance / prevent overfitting. In equation learning (that is, using basis functions to generate the covariates for regression) the situation is completely different regarding these two points: Firstly, the basis function expansion introduces specific and strong correlations, and LASSO therefore fails to identify the true covariates (reproduction). And secondly, regarding prevention of overfitting, the aim of equation learning is not to select some covariates as long as variance is reduced and predictive power is achieved, instead, the goal is to reproduce the ground truth equations which correspond to specific covariates. We included LASSO to make this important point. The same applies to standard SR, although here we demonstrate that the theoretically known model selection properties of the Bayesian model evidence make SR a strong method for equation learning. In fact, the state-of-the-art methods we compare against adapted LASSO (i.e. regularization) and SR for the same reasons.
> The limits of LASSO relevant for equation learning are discussed in:
> Su, W., Bogdan, M., & Candès, E. (2017). False discoveries occur early on the Lasso path. The Annals of Statistics, 45(5), 2133–2150. https://doi.org/10.1214/16-AOS1521,
> which we will add as a reference.
>
> (3) Adding the paragraph on R^2 was in response to previous reviews, we are happy to move it to the appendix.
>
> (4) We agree, "less greedy" might not be ideal, we hopefully come up with a more catchy phrasing.
>
> In response to the requested changes:
>
> (1) We are happy to move the pseudo-code from the appendix to the main text (we can use the space freed up by omitting details on R^2).
>
> (2) Computational complexity in terms of scaling is a little tricky for greedy algorithms. What can be said in general (number of elementary computations in the worst case) is mentioned in the beginning of section 3.4. We are, however, happy to expand on that a little, but these figures will naturally be of limited generality.
>
> (3) We are also happy to expand explanations in our literature review. We agree that our statement "Greedy algorithms have proven to be more successful than LASSO in equation learning" is a little misleading, for example, the relaxed LASSO (SR3) is not a greedy algorithm and has also been shown to be superior to LASSO. This does not make our statement wrong but leaves out a part of the whole picture. The aim of our statement was to promote greedy algorithms as one possible way to overcome the difficulties LASSO has, and we will make it more clear in a revised version. Evidence for our statement can be found in the given references Champion et al., Kaiser et al., Niven et al., Vaddireddy et al., Rudy et al., which we will cite with the statement in a revised version of our manuscript. As mentioned above, this is specific to equation learning and does not hold for linear regression in general.

---

### Review · Reviewer_ihmg · 2023-07-10

**Summary Of Contributions:**

In this paper, the authors propose a not-so-greedy approach for model selection. The algorithm only works when the true model is a linear combination of basis functions and the number of non-zero weights is small.

The authors propose two algorithms: SR and LG. SR first follows a forward procedure to include potential basis functions and then uses a backward step. The LG algorithm, which is the strongest algorithm, only works for polynomials. It first takes all the polynomials of degree one and keeps the best model. To the obtained model, it adds all the polynomials of degree two and keeps the best model.

**Audience:**

No

**Broader Impact Concerns:**

None.

**Claims And Evidence:**

No

**Requested Changes:**

I do not think this paper is interesting to the ML community and no change in it would make me reconsider this decision. For me, this paper should have been an editorial rejection.

**Strengths And Weaknesses:**

The paper has many weaknesses.

1 The proposed algorithm only works in the model class containing the true model. This is hardly the case.

2 The proposed LG algorithm has the same order complexity as a full search. Because the binomial (p p/2) also grows exponentially in p. This algorithm would not work for larger p or a solution that contained many non-zero entries.

3 Lasso is as good as the proposed method for the Lorenz attractor.

4 For the toy data in Section 4.1, I am surprised that LASSO performs so poorly. I have the impression that lambda is set too high. But the authors do not say how they adjust lambda.

5 The R2 is used for model selection. But the authors do not say whether test or training data is used for computing R2. If it is the former, how much data does the test set contain? If it is the latter, how do they avoid overfitting?

6 The authors talk about feature selection. But there are many relevant missing literature studies. For example https://www.jmlr.org/papers/special/feature03.html and https://bayes.wustl.edu/MacKay/pred.pdf.

7 The paper is mostly written in a frequentist manner, but it uses some concepts from Bayesian statistics that do not make sense. For example, the solution for the regression problem uses regularized least squares, but the author proposes a Bayesian model selection procedure.

---

> ### Author Response · Authors · 2023-07-10
> **Response to Reviewer's Summary**
>
> We thank the reviewer for their critical review. Unfortunately, the reviewer seems to have misunderstood the point of our work, and his comments do not apply to our manuscript. Below we explain in detail why we think so.
>
> The main misconception we think is that our paper is not on feature selection as a whole, but on the particular case of equation learning with its own unique challenges and objectives:
>
> All of the equation learning literature will assume that the true model can be built from the analytic elements used in the approach, anything else would defeat the point of equation learning. An approach to equation learning that considers all possible model equations that exist in finite time and with finite resources appears impossible.
>
> Compared to general feature selection, equation learning has two main characteristics, which is why it is not a general feature selection problem: Firstly, the basis function expansion introduces specific and strong correlations, and approaches like LASSO therefore fail to identify the true features. And secondly, the aim of equation learning is not to select features as long as variance is reduced (in order to prevent overfitting) and predictive power is achieved, instead, the goal is to reproduce the ground truth equations which correspond to specific features.
>
> Also, the summary given in the review does not describe our algorithms correctly. SR is bidirectional stepwise regression utilizing the Bayesian model evidence as a criterion to select or deselect basis functions, that is, the method does not stop after one backward step but continues after the criterion has been optimized in the search space. Our contribution here is to demonstrate how well the evidence from an empirical Bayes approach works for equation learning. Our LG algorithm, as well as the proposed SR implementation, works for all basis functions, not only for polynomials as the reviewer states, as the choice of basis function is independent from the methods used. Furthermore, as explained in section 3.5, we cover a vastly larger model space than just the polynomial space used for the function expansion. And finally, the algorithm does more than just keeping the best model for each model size -- please refer, for instance, to our Figure 1 and its caption for a brief explanation.
>
> Response to the listed weaknesses continues in the next comment.

---

> ### Author Response · Authors · 2023-07-10
> **Response to weaknesses stated by Reviewer**
>
> We now address the weakness listed.
>
> 1) We have addressed this point already above. We may add furthermore that the reviewer is of course correct, the true model will never be contained in the model class in a realistic application, as is the case for any other data-driven modelling approach. This is simply because in science one always has to look at an abstraction of reality, otherwise the influences to consider are beyond any computational capacity imaginable. In fact, we would not have noise whatsoever, as noise is just a result of deterministic effects (quantum physics put aside) that is deliberately decided to be ignored. This delicate balance of what to include to model, and what to leave as noise and describe statistically, is at the heart of this work (and basically all of science). We have enabled Bayesian model selection to achieve that delicate balance.
>
> 2) Firstly, the LG algorithm has a stopping criterion, to strike that perfect balance mentioned above in point 1). Secondly, we apply a heuristically developed mildly greedy step each iteration to reduce the model space, which allows us to go to model sizes of p=8 on a standard laptop (note that most model equations in nature have 2 to 4 terms, see e.g. the more than 40 physical laws in our reference Udrescu & Tegmark (2020)); and p=8 does not have to be the limit, should one still wish to go beyond. Our algorithm therefore has not the same order complexity as a full search -- it is in fact our achievement to make it as close to a full search as it seems possible while maintaining feasibility. We also point out that forward selection procedures have a stopping criterion, and that they would typically not for, e.g., deep learning. In fact, any method can be broken if applied to problems it has not been designed for.
>
> 3) LASSO is as good as the proposed methods if we look at the MAE only, which is the least relevant measure. As explained above, the point in equation learning is not to just make good predictions (for this we would use other machine learning methods), but to do so parsimoniously (that is, keeping the model as small as possible), and to uncover true laws from data. Please refer to the first column in Figure 3: while our methods are as good as LASSO (second row), they require only a fraction of terms (third row), and have a very high identification accuracy of true terms, compared to identical zero as for LASSO (first row). This is of course as LASSO was not designed to identify equations from data, which brings us back to the arguments above, only here it is the correlations between features that break LASSO. While this is known in the equation learning literature, it is not often discussed, which is why we included LASSO to make this point.
>
> 4) LASSO performs poorly for the random polynomial test, as the features derived from the basis function expansion are highly correlated. It is known, as discussed in point 3) above, that LASSO does not work well in this case, see, e.g., our reference Champion et al. (2020), and "Su, W., Bogdan, M., & Candès, E. (2017). False discoveries occur early on the Lasso path. The Annals of Statistics, 45(5), 2133–2150. https://doi.org/10.1214/16-AOS1521", which we will add as a reference. The regularization strength lambda is set by the usual 5-fold Cross-Validation (CV), as explained below Table 1 in our manuscript. When we set lambda manually to make LASSO produce the correct model size, we found a deterioration of performance, even though we are using information that is to be inferred from the data. The procedure we used performed best out of all we tried.
>
> 5) Again, the reviewer seems to look at our work from the general machine learning point of view. The splitting into training and test data is not applicable here (apart from using CV to tune the benchmark methods). All of the data is used as training data to learn the model. Then, we solve the learned and the true model and compare the solutions for a large number of randomly selected initial conditions. Overfitting is avoided by the use of the Bayesian model evidence in the case of LG-$p(M)$, and the heuristically developed selection procedure found for LG-$R^2$. Please refer to the description given in the manuscript (e.g. figure 1 or section 3.4). It is one of our main findings how R^2 can still be used for equation learning despite its lack of overfitting penalty, and that it does not seem to be impaired by the high correlations generated by the basis function expansion.
>
> 6) The paper is on equation learning, which is related to feature selection, but most of the usual feature selection methods do not work just like that as explained above. We are still happy to look at the list of references provided to see what is relevant to our work.
>
> 7) The reviewer is correct, we harvest synergy effects by combining various methods in a novel way, which we see as a strength and not as a weakness of our work.

---

> ### Author Response · Authors · 2023-07-10
> **Response to Reviewer -- Conclusion**
>
> From the points raised by the reviewer, we consider including a focused paragraph that highlights the specifics of equation learning within the broader discipline of feature selection, in addition to mentioning the specifics as we develop the methods as it is now. This way, we hope to avoid misunderstandings for readers coming from different disciplines.
>
> We hope we did clarify all points raised such that the reviewer can re-assess our work from a more suitable angle.

---

### Review · Reviewer_F8VX · 2023-07-26

**Summary Of Contributions:**

The authors present a Bayesian version of the system identification problem employing classical notions from statistics. In particular, the authors modify the SINDy method to include a forward-backward stepwise regression algorithm to find the coefficient weights, which emplyed the $R^2$ statistic and approximation of the model evidence under assumptions of normality.

**Audience:**

Yes

**Claims And Evidence:**

No

**Requested Changes:**

S denotes merely stengthens paper, C denotes critical for acceptance.

- S: (Sec 2) Nardini et al are incorrectly cited they use NN to denoise the data and STLSQ for sparse regression. i.e. NNs are not employed for model selection.
- S: (Sec 2) the "improved noise robustness" category should include reference to weak formulations of SINDy (e.g. WSINDy)
- C: (Sec 2) Review of Bayesian-based sparse regression is lacking, e.g. "Robust data-driven discovery of governing physical laws with error bars", Zhang, Lin
- S: Eq 1: for identifying dynamical systems one rarely has an iid normal error structure. The authors should comment on this
- S: (after eq 6, and throughout) it should be made more clear that $k_n(\pmb{x})$ refers to a set of functions
- C: beginning of sec. 3.4, "a non-greedy algorithm would consider...", this is slightly incorrect, convex relaxations of sparse regression problems are not greedy and do not exhaustively consider all possible sparsity patterns. The authors should clarify what is meant by "greedy" vs "non-greedy", as typically sparse regression algorithms that build the solution term by term are referred to as greedy (e.g. OMP), then there are thresholding methods (STLSQ, IHT), and convex relaxations (LASSO). Thresholding is not typically considered a greedy approach as it is here.
- C: (sec 3.4) Considering all ${p \choose m}$ options is still a combinatorially hard problem. For instance, $p=20$, $m=5$ gives ${p \choose m}=15504$. In fact, many sparse regression algorithms assume that $m$ is a user input (see OMP, COSAMP). Since this part of the algorithm will not scale well, the authors should comment on how to adapt it to more realistic settings. Is the "additional pruning step" meant to circumvent this? If so more explanation is needed. How many iterations of not being selected are allowed before the term is omitted completely? Can it ever be regained?
- C: Compared to the algorithms in the appendix (which should be listed in the main body) section 3.4 is lacking key details. In contrast, Appendix C is missing any mention of this key "pruning step", so there appears to be a mismatch between the algorithm as stated in the body and in the appendix.
- S: much of appendix A is a repeat of the manuscript
- S: the switch from __y__ to $y$ in eq. 18-22 is unmotivated
- S: "is essentially is" after eq 22
- C: after eq 25, "we see that the evidence p(y) is in fact the model-likelihood p(y|M)" this is not clear to me, the authors should explain why p(y) equals p(y|M). In fact, if this is true then the posterior equals the prior always.
- C: after eq 25, in general the authors should define $\mathcal{M}$. Is $\mathcal{M}$ a set of terms, so an indicator vector in {0,1}$^p$ (equal to the sparsity pattern of $\pmb{w}$)? Or is it the entire set of possible functions that are used in the regression? Is it a function space? A "constant prior" $p_{pr}(\mathcal{M})$ may not make sense in some of these interpretations.
- S: eq 29-31 are repeated in the line above, authors should pick one.
- S: after eq 31, "...is $M$ is..."
- S: much of the appendix should be given in the rest of the paper, for instance the discussion in eq 32-35 etc.
- C: eq 35 and mention of finite differences throughout should include mention of methods to circumvent the errors of this approach (namely using a weak formulation)
- S: "The problem is that basis functions like $1/(1+x^n)$ are divergent for certain values of x, and are also quite limiting in their form", perhaps the authors should say "certain values of $n$" since $n$ even leads to bounded functions. While this notion is interesting, I would caution away from claiming these functions are "limiting" (they appear in many biochemical models)
- S: "needs to be determined from a 1-dimensional numerical root-finding algorithm" this is interesting, could the authors provide the details? Conditioning would seem to be a concern here.
- S: $y^T$ should be bold in equation 45

**Strengths And Weaknesses:**

The strengths include

(i) clarity and conciseness of exposition \
(ii) an interesting new angle of incorporating the Bayesian evidence.

The weaknesses include

(i) repetition of text between the manuscript and appendix \
(ii) incomplete or naive arguments (e.g. the authors claim to reduce the complexity of the sparse regression problem by only considering all models of sparsity $m<p$, so considering ${p \choose m}$ total models, instead of all $2^p$ models, when the former is still NP-hard) \
(iii) While it is useful to identify $R^2$ as a criterion, this is by no means a new approach, and the authors heavily rely on it for their method, reducing the novelty of the manuscript.

---

> ### Author Response · Authors · 2023-08-02
> **Response to weaknesses**
>
> We thank the reviewer for their detailed report and appreciate the constructive suggestions. We in particular welcome that our effort to present our work in a clear and concise way has been noted, and that it is considered a new and interesting angle of incorporating Bayesian model selection in equation learning.
>
> Below, we respond to the weaknesses stated by the reviewer.
>
> (i) repetition of text between the manuscript and appendix.
>
> Reply:
>
> We did this purposefully, to have a self-contained introduction to the field/approach in the appendix, as we found that this was beneficial to readers coming from neighbouring fields, and to avoid having to jump between main text and appendix. We would not mind having a more detailed introduction in the main text (dropping the corresponding parts in the appendix). However, this was again noted as a weakness by readers already familiar with equation learning, as it takes away the focus from our results and discussion. In the end, as a compromise, we decided to put a detailed version of the theory and methods in the appendix, thus also meeting the page limit of 12, and still accommodating a broader audience.
>
> (ii) incomplete or naive arguments (e.g. the authors claim to reduce the complexity of the sparse regression problem by only considering all models of sparsity $m<p$, so considering ${p \choose m}$ total models, instead of all $2^p$ models, when the former is still NP-hard).
>
> Reply:
>
> We see the point of the reviewer in the given example. We did not mean to say we change the complexity class of the problem, it is of course still NP. We used ''complexity'' in a broader sense (like ``extent''), that is breaking down the set of all models into smaller sets such that a near-comprehensive search becomes possible with our heuristic. We will improve this statement in the revised version and also carefully look out for other such examples in our manuscript.\\
>
> (iii) While it is useful to identify $R^2$ as a criterion, this is by no means a new approach, and the authors heavily rely on it for their method, reducing the novelty of the manuscript.
>
> Reply:
>
> We see our contribution in demonstrating how such a simple and cheap criterion like $R^2$ can enable a near-comprehensive search in a vast model space. We do of course not claim that the simple use of $R^2$ in statistical inference is novel, but it is the way we exploit its properties and combine it with other criteria to tackle the challenges of equation learning that we have seen in the literature, and which would ultimately allow employing even computationally costly methods like model evidence estimation.
>
> As a side note, as it comes in the reviewer's suggestions: Incorporating general Bayesian model selection (not limited to conjugate priors) would even allow using any distribution for the error term, opening up a more probabilistic approach to equation learning. This will be taken up in future work.
>
> Regarding the statement that we heavily rely on identifying $R^2$ as a criterion: Besides the use of $R^2$ (for a pre-selection step!), we also demonstrate how bi-directional stepwise regression equipped with the model evidence (SR) exhibits a particularly high identification accuracy. In total, we propose and investigate three new heuristics for equation learning: LG-$R^2$ using only $R^2$ (in a way that, surprisingly, incorporates a sharp complexity penalty), LG-$p(\mathcal{M})$ mainly employing the model evidence and building on $R^2$ as a pre-step, and SR using only the model evidence -- all of which build upon existing methods from statistical inference (like most of equation learning methods, in fact), but, to our knowledge, none of these has been tailored to equation learning and been investigated in a simulation study.

---

> ### Author Response · Authors · 2023-08-02
> **Response to comments, part 1**
>
> We now address the list of requested changes, where we limit ourselves to the critical points (the non-critical points are suggestions that we are happy to implement).
>
> Comment:
>
> (Sec 2) Review of Bayesian-based sparse regression is lacking, e.g. "Robust data-driven discovery of governing physical laws with error bars", Zhang, Lin.
>
> Reply:
>
> We agree that a more Bayesian focus in the literature review is appropriate, and we will add the mentioned reference among others in the revised version.
>
> Comment:
>
> beginning of sec. 3.4, "a non-greedy algorithm would consider...", this is slightly incorrect, convex relaxations of sparse regression problems are not greedy and do not exhaustively consider all possible sparsity patterns. The authors should clarify what is meant by "greedy" vs "non-greedy", as typically sparse regression algorithms that build the solution term by term are referred to as greedy (e.g. OMP), then there are thresholding methods (STLSQ, IHT), and convex relaxations (LASSO). Thresholding is not typically considered a greedy approach as it is here.
>
> Reply:
>
> We thank the reviewer for pointing this out. We indeed understand greedy algorithms as a broader class than the reviewer into which we also count thresholding methods as a sub-class, as these methods still feature the characteristic ``step and selection'' procedure. And we did not consider convex optimisation as part of the greedy versus non-greedy distinction at all. We do see the benefit in distinguishing greedy and thresholding methods in terms of clarity and consistency, and defining what we mean by greedy and non-greedy, and we will adapt this paragraph in the revised version accordingly.
>
> Comment:
>
> (sec 3.4) Considering all ${p \choose m}$ options is still a combinatorially hard problem. For instance, $p=20$, $m=5$ gives ${p \choose m}=15504$. In fact, many sparse regression algorithms assume that $m$ is a user input (see OMP, COSAMP). Since this part of the algorithm will not scale well, the authors should comment on how to adapt it to more realistic settings. Is the "additional pruning step" meant to circumvent this? If so more explanation is needed. How many iterations of not being selected are allowed before the term is omitted completely? Can it ever be regained?
>
> Reply:
>
> Indeed, considering ${p \choose m}$ models is still a hard problem, and our contribution is that we can still tackle that problem by exploiting the fact that $R^2$ is extremely cheap to compute for standardised data and moderate $m\lesssim 10$ (and we are interested in moderate $m$ anyway in the context of equation learning). In a previous version, we did not use the pruning step, which only becomes important for $m\gtrsim 5$. That is to say, the main reason that we can deal with the remaining combinatorial problem is the use of $R^2$ the way we do, the pruning step just extends feasibility a little further to larger models. For the benefit of the reviewer, we timed our $R^2$ procedure without pruning on the example given, that is $p=20$ and $m=5$, and it took about $0.8$\,s to evaluate all $15504$ models for $N=1000$ datapoints on a standard laptop. We also observe a near-linear scaling per model in $N$ and $p$. Consequently, while the scaling of ${p \choose m}$ means the method becomes infeasible for large models, it is still very well suited for equation learning where we typically have $2$-$4$ terms in the equations, and we can still manage $8$ terms without problem even on a standard laptop.
> As this has also been commented on by the other reviewers, we will include a short complexity/runtime analysis of using $R^2$ in this way in the revised version, to give a better insight into the computational requirements, and demonstrate the feasibility.
> We also agree that we skipped too many details in section 3.4 (still having the details in the appendix though), and will change that in the revised version, including answering the questions of the reviewer. For the reviewer's benefit: We found that two iterations for omitting a non-selected term are a universally good choice for speed-up without having to compromise in identification accuracy. After being omitted, the term cannot be regained in later iterations.

---

> ### Author Response · Authors · 2023-08-02
> **Response to comments, part 2**
>
> Comment:
>
> Compared to the algorithms in the appendix (which should be listed in the main body) section 3.4 is lacking key details. In contrast, Appendix C is missing any mention of this key "pruning step", so there appears to be a mismatch between the algorithm as stated in the body and in the appendix.
>
> Reply:
>
> The pruning step is in fact included in Algorithm 3 (lines 14-15), and also explained in the text on page 22 (''As for larger values of $r$ the number of candidate models can easily reach hundreds of millions, ...'').
>
> In response to the main text lacking key details: We meant to focus in the main text on details that are important for understanding our procedures, putting details that are important for reproduction into the appendix. In line with our reply to the previous point, we will include more details in the main text in the revised version. We will also move the (main) algorithms back into the main text (we had these in the main text before).
>
> Comment:
>
> after eq 25, ''we see that the evidence $p(y)$ is in fact the model-likelihood $p(y|\mathcal{M})$'' this is not clear to me, the authors should explain why $p(y)$ equals $p(y|\mathcal{M})$. In fact, if this is true then the posterior equals the prior always.
>
> Reply:
>
> The notation in eqs (24) and (25) is indeed not very clear, as $p(y)$ in eq (24) with the added conditioning on $\mathcal{M}$ is not written down before eq (25), and therefore it looks like the $p(y)$ in both equations are the same quantity, which they are in fact not.
>
> To clarify: Going from eq (24) to eq (25), we make the silent conditioning on $\mathcal{M}$ explicit in the likelihood and in the evidence in the denominator, such that $p(y)$ becomes $p(y|\mathcal{M})$. We then reinterpret $p(y|\mathcal{M})$ as a likelihood that only depends on the choice of model $\mathcal{M}$, that is, where the parameters have been marginalised out. This is meant by our statement quoted in the reviewer's comment. We can then apply Bayes theorem again to reverse the conditioning of $p(y|\mathcal{M})$ and arrive at eq (25). The denominator $p(y)$ in eq (25) is now only a theoretical construction, as it would involve a marginalisation of all models (we could, of course, define a sigma-Algebra from a set of models which would in principle allow the computation of $p(y)$). However, as $p(y)$ is only a constant in terms of $\mathcal{M}$, we can ignore it, and see that maximising the evidence corresponds to finding the most likely model given the data as stated below (25) (if, for simplicity of argument, models are a priori assumed to be equally likely).
>
> We will of course improve the clarity of the paragraph in the revised version, adding the steps explained above.
>
> Comment:
>
> after eq 25, in general the authors should define $\mathcal{M}$. Is $\mathcal{M}$ a set of terms, so an indicator vector in $\\\{0,1\\\}^p$ (equal to the sparsity pattern of $\boldsymbol{w}$)? Or is it the entire set of possible functions that are used in the regression? Is it a function space? A ''constant prior'' $p_{pr}(\mathcal{M})$ may not make sense in some of these interpretations.
>
> Reply:
>
> As written above eq (25), $\mathcal{M}$ is defined as the representation by the basis function expansion, but admittedly, this might be misleading. Generally, the model $\mathcal{M}$ defines the likelihood in eq (24). Here, the likelihood is only fixed by the sparsity pattern of $\boldsymbol{w}$, that is, each $\mathcal{M}$ is uniquely defined by an indicator vector $v\in \\\{0,1\\\}^p$. A uniform (constant, uninformative) prior makes sense in this setting as we have (in general) no information on which model to favour over other models before we have seen the data (unless we test models by analytical means without specifying the parameters, which would be possible in principle, but appears infeasible without developing a clever test method which would be a different project altogether).
>
> Comment:
>
> eq 35 and mention of finite differences throughout should include mention of methods to circumvent the errors of this approach (namely using a weak formulation).
>
> Reply:
>
> We will add a comment about the possibility of using a weak formulation in the revised version.\\
>
> Overall:
>
> We are confident that we resolved all (critical) points raised by the reviewer, and are happy to discuss any point further should it be necessary. We agree to all other (non-critical) points not addressed above and will make the appropriate changes in the revised version of the manuscript. We thank the reviewer again, we feel our manuscript improves considerably from their feedback.

---

### Decision · Action_Editor_QMah · 2023-10-17

**Recommendation:** Accept with minor revision

**Comment:**

The reviewers were a bit split: one was tentatively positive, another found merit in the subfield of equation learning but had many comments that they felt were most suited for a major revision, and the third found the assumptions of equation learning quite restrictive (i.e., that the true model is in the class, and related to this, that the residual was iid Gaussian).

I read the paper myself, keeping the TMLR [evaluation criteria](https://jmlr.org/tmlr/editorial-policies.html) in mind:
- *Are the claims made in the submission supported by accurate, convincing and clear evidence?*
- *Would at least some individuals in TMLR's audience be interested in knowing the findings of this paper?*

*Papers should be accepted if they meet the criteria, even if the contribution or significance of the work is modest*.

Valid grounds for rejection are lack of evidence, incorrectly claimed novelty, and re-implementing an idea that has been reproduced before.  I didn't see any of these issues.

Under these basic criteria, I find the paper does pass the threshold.  This is a new idea, and there is numerical evidence to substantiate some of the claims.

However, the reviewers raised good points, and in particular reviewer F8VX requested significant changes.  Hence I am recommending an accept-with-revisions.
- Please address reviewer F8VX's comments
- There are some formatting issues, for example `\citet` and `\citep` are mixed up for many of the references.

**Audience:**

This is for the equation-learning subcommunity, which is relatively small but growing. One reviewer implicitly expressed concern about the size of the audience, but the other reviewers did not. I think there are some individuals in the TMLR audience that would be interested.

**Claims And Evidence:**

The authors propose 3 new methods for solving equation learning problems (a new field that's not quite system identification, not quite sparse regression, but somewhere in between).  The new methods are heuristics, just as many of the existing state-of-the-art methods are, and I do not see that as an inherent weakness. The new methods can have very high computational complexity (as some of the reviewers were concerned about), i.e., in all regimes, this does not beat brute-force enumeration. But in some regimes (e.g., m <= 8), this method is tractable, and better than brute-force.

The main claims are that these methods can select better models than other solvers, and that it is efficient in appropriate regimes.

The main evidence for accuracy is numerical experiments. These are reasonable convincing.  The main evidence for the efficiency is the fact that the algorithms were about to run for the numerical experiments, and the derived time complexity expression for given steps (e.g., roughly (p choose m) for some of the steps).

---

> ### Author Response · Authors · 2023-11-20
> **Camera-ready version submitted**
>
> Dear Stephen Becker,
> we thank you for your time devoted to our work. We also thank the reviewers and other editors involved.
> We have submitted the camera-ready version (within the deadline), where we addressed all remaining issues. If there is anything else to do, please do let us know.
> Best wishes,
> the authors